# A Mean-Field Game Approach to Cloud Resource Management with Function Approximation

**Weichao Mao**
University of Illinois Urbana-Champaign
weichao2@illinois.edu

**Haoran Qiu**
University of Illinois Urbana-Champaign
haoranq4@illinois.edu

**Chen Wang**
IBM Research
chen.wang1@ibm.com

**Hubertus Franke**
IBM Research
frankeh@us.ibm.com

**Zbigniew Kalbarczyk**
University of Illinois Urbana-Champaign
kalbarcz@illinois.edu

**Ravishankar K. Iyer**
University of Illinois Urbana-Champaign
rkiyer@illinois.edu

**Tamer Başar**
University of Illinois Urbana-Champaign
basar1@illinois.edu

## Abstract

Reinforcement learning (RL) has gained increasing popularity for resource management in cloud services such as serverless computing. As self-interested users compete for shared resources in a cluster, the multi-tenancy nature of serverless platforms necessitates multi-agent reinforcement learning (MARL) solutions, which often suffer from severe scalability issues. In this paper, we propose a mean-field game (MFG) approach to cloud resource management that is scalable to a large number of users and applications and incorporates function approximation to deal with the large state-action spaces in real-world serverless platforms. Specifically, we present an online natural actor-critic algorithm for learning in MFGs compatible with various forms of function approximation. We theoretically establish its finite-time convergence to the regularized Nash equilibrium under linear function approximation and softmax parameterization. We further implement our algorithm using both linear and neural-network function approximations, and evaluate our solution on an open-source serverless platform, OpenWhisk, with real-world workloads from production traces. Experimental results demonstrate that our approach is scalable to a large number of users and significantly outperforms various baselines in terms of function latency and resource utilization efficiency.

## 1 Introduction

Serverless computing[1] is an emerging cloud computing paradigm that allows users to develop and run their applications without worrying about the configurations of the infrastructure (containers or virtual machines) [7, 33]. In contrast to traditional cloud computing, serverless users are only charged by the function execution time and their operational cost is reduced since the cloud provider takes care of the resource management (e.g., serverless function provisioning and scheduling) on their behalf. The tension between cloud providers and users incurs a more complex resource management problem: The platform needs to meet various user-defined serverless function quality-of-service (QoS) guarantees while the provider aims to keep the cluster resource utilization at a high level in the face of elastic and bursty function requests. Numerous heuristics-based solutions (e.g., [64, 67, 69, 77]) have been

---

[1]We focus on Function-as-a-Service (FaaS), which is the most common serverless service in cloud computing.

36th Conference on Neural Information Processing Systems (NeurIPS 2022).

utilized, yet they inevitably rely on extensive system-specific domain knowledge and painstaking tuning for specific workloads.

Recently, reinforcement learning (RL) based approaches have attracted increasing attention for dynamic resource management as RL helps automatically adapt to a specific user workload. Each function is managed by an RL agent, leaving human operators out of the loop. The fact that large-scale cloud computing platforms have many users with independent (and sometimes conflicting) requirements adds another layer of complication for serverless function resource management. Since functions from multiple self-interested users coexist and compete for shared resources in a cluster, cloud providers further need to take into account the potential strategic behavior of the users. The multi-agent reinforcement learning (MARL) paradigm [39] is especially well-suited here, as it naturally integrates game-theoretical thinking into the sequential decision-making problems of multi-agent systems. However, a well-known challenge in multi-agent reinforcement learning is *scalability*, as many MARL algorithms suffer from exponential computation & sample complexity in the number of agents, a phenomenon known as "the curse of multiagents" [32, 65, 41, 19, 42, 18]. The scalability bottleneck needs to be properly addressed before applying MARL to real-world cloud serverless platforms, which typically involve a large number of users and functions in a cluster [1].

To circumvent the scalability challenge, we resort to the mean-field approximation [37, 30] and propose a scalable mean-field game (MFG) approach to cloud resource management. The underlying principle of MFGs is to approximate the finite-agent game with an infinite-population limit, where each agent's influence on the overall system becomes infinitesimal. In an MFG with infinite homogeneous agents, the collective behavior of all the agents can be effectively summarized as a population distribution, which is usually specified as the empirical distribution of the agents' states. Such an approximation leads to a tractable solution to the otherwise challenging MARL problem, as each agent no longer needs to keep track of the historical behavior of every other agent to reason about their internal information, which typically incurs a prohibitive combinatorial complexity. Existing theory [54] also shows that the approximation scheme does not lose much of optimality when applying the policies learned from the infinite-agent game back to the original finite-population game.

However, the convergence of learning algorithms in MFGs has been chiefly established either in the tabular setting with small state & action spaces [66, 28], or in the linear-quadratic case with structural assumptions on the system dynamics [23, 73]. Few results have considered *function approximation* in mean-field games with large (and even infinite) state & action spaces, which is the typical case for many real-world application scenarios including cloud computing. In this paper, motivated by the large state & action spaces in cloud resource management problems, we make an initial attempt to understand the effects of function approximation in MFGs with generic system dynamics. While we believe that our proposed methodology is general enough to be applicable to a wide range of scenarios, we demonstrate the effectiveness of our approach in a real-world cloud resource management problem as an important application domain.

**Contributions.** Our contribution is threefold: (1) Algorithmically, we propose a natural actor-critic (NAC) learning paradigm for MFGs that is compatible with various forms of function approximations. Our method is particularly practical due to an *online* property [78], in the sense that it need not fix the mean-field state to calculate the best response at each iteration, but instead let the mean-field naturally evolve as the agents learn. (2) Theoretically, we establish the finite-time convergence of NAC with linear function approximation and softmax parameterization as a critical stepping stone. We prove that the resulting algorithm converges to the regularized Nash equilibrium (NE) of the MFG at an $\widetilde{O}(T^{-1/5})$ rate. (3) Empirically, we evaluate NAC on classic MFG benchmarks and demonstrate its convergence behavior. As an important motivating example and test bench, we also incorporate a practical variant of NAC (with both linear and neural-network function approximations) into the resource management module of an open-source serverless platform, OpenWhisk [22]. Our experimental results on real-world production workloads show that NAC is scalable to a large number of agents in serverless resource management and significantly outperforms various baselines in terms of both function latency and resource utilization.

**Related Work.** Mean-field games have been introduced by [37] and [30] to study continuous-time differential games with infinite identical agents. For discrete-time MFGs, the existence and/or uniqueness of the Nash equilibrium have been studied in [26, 70, 45, 54]. The mean-field regime has been investigated under various settings, including games with linear-quadratic structures [9, 72], major and minor agents [47], partial observability [55], risk-sensitivity [56], and so on. When

the environment dynamics are unknown, learning-based algorithms are commonly used to learn the NE from data [80, 73, 23, 28, 20, 49, 5, 27, 16], but most of these works rely on a "double-loop" fixed-point iterative process that can lead to heavy reward fluctuations and are less practical. Our "single-loop" algorithm is most related to [78, 66, 48, 74], but they do not consider function approximation to deal with large state and action spaces, and [66] and [48] only yield asymptotic convergence guarantees. On the application side, learning algorithms in MFGs have been applied to economics [6] and animal behavior simulation [50], among others, while in this paper, we identify that large-scale cloud resource management is also an ideal application domain.

Our work is also related to policy optimization in single-agent RL, especially natural policy gradient (NPG) methods [35]. For the tabular setting, the (global) convergence of NPG has been established in a series of works [63, 2, 43, 79, 14]. In particular, [2] has proved that unregularized NPG with softmax parameterization achieves an $O(1/T)$ convergence rate, while [14] has further strengthened the result by showing linear convergence of NPG when equipped with entropy regularization. In the regime of function approximation, [2] has shown that NPG with linear function approximation and softmax parameterization attains a $O(1/\sqrt{T})$ convergence rate subject to some function approximation error. Our results are largely built upon [13], which has established a linear convergence when further exploiting entropy regularization. In particular, a standard "double-loop" mean-field approach will lead to a setting very similar to that of [13] in the inner loop, because by fixing the population distribution, the learning agent effectively faces a single-agent problem. However, since such double-loop solutions are hardly practical, we instead consider an online setting where the environment simultaneously evolves as the agents update their policies. The environmental non-stationarity adds another layer of complexity and makes [13] not directly applicable. Going beyond linear function approximation, [76] has investigated NPG with over-parameterized two-layer neural networks, a more powerful approximation scheme that is not theoretically pursued but only empirically evaluated in our work. It is also worth mentioning that many successful empirical algorithms, such as TRPO [58] and PPO [59], can be considered as natural variants of NPG.

RL-based resource management approaches [8, 36, 40, 51, 57, 82] have been recently proposed to achieve efficient resource utilization and application quality-of-service (QoS), which outperform baseline rule-based methods. For instance, FIRM [51] is an RL-based resource management framework for cloud microservices to tackle the under-utilization issue and QoS violations. [57] has proposed a Q-learning-based autoscaler that decides the horizontal concurrency for a serverless function. FaaSRank [82] is an RL-based serverless function scheduler that uses PPO [59] to assign function requests to available servers, yet both [57] and [82] only consider the objective of minimizing the function latency. Finally, existing works only consider single-agent RL approaches being trained and evaluated in an isolated environment, while the cloud is multi-tenant environment by nature.

## 2 Preliminaries

We first consider a classic Markov game with $N$ agents. Each agent has a state space $\mathcal{S}$ and an action space $\mathcal{A}$. At each time step $t$, the state of agent $i \in [N]$ is denoted by $s_t^i \in \mathcal{S}$, and the agent takes an action $a_t^i \in \mathcal{A}$ according to a certain policy. Given the current state profile $\boldsymbol{s}_t = (s_t^1, \ldots, s_t^N)$, agent $i$ receives a reward determined by a reward function $\tilde{r}^i(\boldsymbol{s}_t, a_t^i)$, and transitions to a new state $s_{t+1}^i \sim \tilde{P}^i(\cdot \mid \boldsymbol{s}_t, a_t^i)$ according to a transition function $\tilde{P}^i$. The goal of each agent is to find a policy that maximizes its expected cumulative reward over time. When $N$ is large, learning in such an $N$-agent Markov game with generic reward structure is known to be notoriously hard [65, 18]. Mean-field games [37, 30], on the other hand, can be viewed as an infinite-population limit ($N \rightarrow \infty$) of the finite-agent game, and provide a tractable approximation to the otherwise challenging problem.

A discrete-time mean-field game (MFG) considers an infinite-number of identical agents [37, 30]. The collective behavior of all the agents is described by population distributon $\mu_t \in \Delta(\mathcal{S})$, also termed a mean-field state, which in practice can be interpreted as the limit of the empirical state distribution, i.e., $\mu_t = \lim_{N \rightarrow \infty} \frac{1}{N} \sum_{i=1}^{N} \delta_{s_t^i}$, where $\delta_s \in \Delta(\mathcal{S})$ is the Dirac measure at $s$. Due to the homogeneity of the agents, we focus on a single representative agent. At each time $t$, the (representative) agent's state is denoted by $s_t \in \mathcal{S}$, and the mean-field state $\mu_t$ describes the probability distribution of $s_t$. Upon taking an action $a_t \in \mathcal{A}$ at $s_t$, the agent receives a reward $r(s_t, a_t, \mu_t)$, and transitions to a new state $s_{t+1} \sim P(\cdot \mid s_t, a_t, \mu_t)$, where $r : \mathcal{S} \times \mathcal{A} \times \Delta(\mathcal{S}) \rightarrow [0, 1]$ is the reward function and $P : \mathcal{S} \times \mathcal{A} \times \Delta(\mathcal{S}) \rightarrow \Delta(\mathcal{S})$ is the state transition function. A (Markov) policy $\pi : \mathcal{S} \rightarrow \Delta(\mathcal{A})$

for the agent is a mapping from the state space to a distribution over the action space, and we use $\Pi$ to denote the set of all Markov policies. Given a population distribution flow $\boldsymbol{\mu} = (\mu_t)_{t \geq 0}$, we define the *value function* of a policy $\pi$ as $V^\pi_{\boldsymbol{\mu}}(s) \stackrel{\text{def}}{=} \mathbb{E}\left[\sum_{t=0}^\infty \gamma^t r(s_t, a_t, \mu_t) \mid s_0 = s\right]$, where $a_t \sim \pi(\cdot \mid s_t), s_{t+1} \sim P(\cdot \mid s_t, a_t, \mu_t)$, and $\gamma \in (0, 1)$ is the discount factor. In the special case of a time-invariant mean-field, i.e., $\mu_t = \mu, \forall t \geq 0$, we slightly abuse the notation and write $V^\pi_{\boldsymbol{\mu}}(s)$ as $V^\pi_\mu(s)$. For an initial state distribution $\rho \in \Delta(\mathcal{S})$, we define the (discounted) state *visitation distribution* as $d^\pi_\mu(s) \stackrel{\text{def}}{=} (1 - \gamma) \sum_{t=0}^\infty \gamma^t \mathbb{P}(s_t = s \mid s_0 \sim \rho)$, where $\mathbb{P}(s_t = s \mid s_0 \sim \rho)$ is the probability that the state $s$ is visited at the $t$-th time step under policy $\pi$ and mean-field $\mu$.

**Entropy Regularization:** We further consider an entropy-regularized value function by augmenting the standard reward objective with an entropy term of the policy. For a fixed mean-field state $\mu$, define

$$V^{\pi,\lambda}_\mu(s) \stackrel{\text{def}}{=} V^\pi_\mu(s) + \lambda H^\pi_\mu(s) = \mathbb{E}\left[\sum_{t=0}^\infty \gamma^t \left(r(s_t, a_t, \mu) - \lambda \log \pi(a_t \mid s_t)\right) \mid s_0 = s\right],$$

where $\lambda > 0$ is a parameter that controls the level of regularization, $H^\pi_\mu(s) \stackrel{\text{def}}{=} \mathbb{E}\left[\sum_{t=0}^\infty \gamma^t \mathcal{H}(\pi(\cdot|s_t)) \mid s_0 = s\right]$, and $\mathcal{H}(\pi(\cdot|s)) = -\sum_{a \in \mathcal{A}} \pi(a|s) \log \pi(a|s)$ is the Shannon entropy. Entropy regularization has been commonly used to encourage exploration and avoid premature convergence to sub-optimal near-deterministic policies [29, 3]. In mean-field games, [5] has also shown that with regularization, the uniqueness of NE is guaranteed under milder assumptions than the unregularized case. We define the soft Q-function and shifted Q-function, respectively, as

$$Q^{\pi,\lambda}_\mu(s, a) \stackrel{\text{def}}{=} r(s, a, \mu) + \gamma \mathbb{E}_{s' \sim P(\cdot|s,a,\mu)}\left[V^{\pi,\lambda}_\mu(s')\right], \text{ and } q^{\pi,\lambda}_\mu(s, a) = Q^{\pi,\lambda}_\mu(s, a) - \lambda \log \pi(a|s),$$

which are related to $V^{\pi,\lambda}_\mu$ in the sense that $V^{\pi,\lambda}_\mu(s) = \mathbb{E}_{a \sim \pi(\cdot|s)}[q^{\pi,\lambda}_\mu(s, a)]$. For a distribution $\rho \in \Delta(\mathcal{S})$, with a slight abuse of notation, we write $V^{\pi,\lambda}_\mu(\rho) = \sum_{s \in \mathcal{S}} \rho(s) V^{\pi,\lambda}_\mu(s)$.

**Policy Parameterization:** We consider parametric policy classes. For each state-action pair $(s, a)$, suppose there exists a $d$-dimensional feature mapping $\phi_{s,a} \in \mathbb{R}^d$, such that $\|\phi_{s,a}\|_2 \leq 1, \forall s \in \mathcal{S}, a \in \mathcal{A}$. A commonly used policy class is softmax parameterization of the form

$$\widetilde{\Pi} = \left\{\pi_\theta(a|s) = \frac{\exp\left(f_\theta(s, a)\right)}{\sum_{a' \in \mathcal{A}} \exp\left(f_\theta(s, a')\right)} : \theta \in \mathbb{R}^d\right\},$$

where $\theta$ is a $d$-dimensional vector that parameterizes the policy, and $f_\theta$ is a differentiable function that can be typically instantiated as a linear function or a neural network. In Sections 3 and 4, we focus on "log-linear" policies that use linear function approximation, where $f_\theta$ takes the specific linear form of $f_\theta(s, a) = \theta^\top \phi_{s,a}, \forall (s, a) \in \mathcal{S} \times \mathcal{A}$. We will also instantiate $f_\theta$ using neural networks later in Section 5. Function approximation helps deal with large state and action spaces, as the feature dimension is usually much smaller, i.e., $d \ll |\mathcal{S}||\mathcal{A}|$, where $|\mathcal{S}|$ can even be infinite in practice. It is worth noting that a parametric policy class may not contain all Markov policies, i.e., $\widetilde{\Pi} \subset \Pi$. Hence, we seek to do as well as the best policy in this class and obtain agnostic results.

**Single-agent Policy Optimization.** Given a fixed mean-field state $\mu$, the representative agent faces a single-agent policy optimization problem: $\max_{\pi \in \Pi} V^{\pi,\lambda}_\mu(s)$, which is equivalent to finding the (entropy-regularized) optimal policy for a single-agent Markov decision process (MDP) induced by $\mu$. It has been shown that the optimal policy is unique whenever $\lambda > 0$ [25]. Hence, we can use $\pi^{\star,\lambda}_\mu$ to denote the (unique) optimal solution to the optimization problem, and define a mapping $\Gamma^\lambda_1 : \Delta(\mathcal{S}) \to \Pi$ such that $\Gamma^\lambda_1(\mu) = \pi^{\star,\lambda}_\mu$. We refer to $\Gamma^\lambda_1$ as the *policy optimization operator*, which maps a mean-field state $\mu$ to the optimal policy $\pi^{\star,\lambda}_\mu$ of the induced MDP.

**Mean-field Dynamics.** Since all agents follow the same policy $\pi$, we can define another mapping $\Gamma_2 : \Pi \times \Delta(\mathcal{S}) \to \Delta(\mathcal{S})$ to describe the evolution of the mean-field. Specifically, the *mean-field dynamics operator* $\Gamma_2$ is defined by $\Gamma_2(\pi, \mu) = \mu^+$, where

$$\mu^+(\cdot) = \int_{\mathcal{S} \times \mathcal{A}} P(\cdot|s, a, \mu)\mu(s)\pi(a|s) \, \mathrm{d}a \, \mathrm{d}s.$$

Intuitively, $\Gamma_2$ characterizes the next mean-field state, given the current mean-field state and the current policy adopted by all the agents. For notational convenience, we further introduce a composite mapping $\Lambda^\lambda : \Delta(\mathcal{S}) \to \Delta(\mathcal{S})$ as $\Lambda^\lambda(\mu) = \Gamma_2(\Gamma^\lambda_1(\mu), \mu)$, which simply combines $\Gamma^\lambda_1$ and $\Gamma_2$.

**Mean-field Equilibrium.** In the following, we introduce the main learning objective of the paper.

**Algorithm 1:** Natural Actor-Critic for MFGs with Linear Function Approximation

---

**1** **Input:** Initial mean-field state $\mu_0$, and regularization level parameter $\lambda$;
**2** Initialize $\theta_0 \leftarrow \mathbf{0}$;
**3** **for** *iteration* $t \leftarrow 0$ *to* $T$ **do**
**4**     **Policy evaluation:** Calculate an $\varepsilon_{\text{critic}}$-accurate estimate $\hat{q}_t^\lambda$ of $q_{\mu_t}^{\pi_t,\lambda}$ that satisfies Equation (2) using, e.g. Algorithm 2;
**5**     **Gradient estimation:** Use $\hat{q}_t^\lambda$ to calculate an $\varepsilon_{\text{actor}}$-accurate estimate $\hat{w}_t$ of the gradient (using, e.g., Algorithm 3) such that $\|\hat{w}_t\|_2 \leq R$ for some $R > 0$, and Equation (3) holds;
**6**     **Mean-field update:** $\mu_{t+1} \leftarrow (1 - \beta_t)\mu_t + \beta_t\Gamma_2(\pi_t, \mu_t)$, where $\beta_t = O(T^{-4/5})$;
**7**     **Policy update:** $\theta_{t+1} \leftarrow \theta_t + \eta_t g_t$, where $g_t = \hat{w}_t - \lambda\theta_t$, and $\eta_t = O(T^{-2/5})/\lambda$;

---

**Definition 1.** *A policy-population pair $(\pi^\star, \mu^\star) \in \Pi \times \Delta(\mathcal{S})$ is a stationary (time-independent) entropy-regularized Nash equilibrium for the mean-field game if it satisfies: (1) Rationality: $\pi^\star = \Gamma_1^\lambda(\mu^\star)$, and (2) consistency: $\mu^\star = \Gamma_2(\pi^\star, \mu^\star)$.*

When $\lambda = 0$, the above definition reduces to that of the standard (unregularized) NE in MFGs [54, 28]. For $\lambda > 0$, the regularized NE $(\pi^\star, \mu^\star)$ also serves as a good approximation of the unregularized one, as characterized by the following error bound [14] (established using the fact that $H_\mu^\pi(s) \leq \frac{\log|\mathcal{A}|}{1-\gamma}$):

$$V_{\mu^\star}^{\pi^\star}(\rho) \leq \max_{\pi \in \Pi} V_{\mu^\star}^\pi(\rho) \leq V_{\mu^\star}^{\pi^\star}(\rho) + \frac{\lambda\log|\mathcal{A}|}{1-\gamma}. \tag{1}$$

When the composite mapping $\Lambda^\lambda$ is a contraction, one can show (using a standard Banach-fixed point theorem) that the regularized NE exists and is unique [5]. Accordingly, the policy-population pairs $\{(\pi_t, \mu_t) : t \geq 0\}$ given by the simple iterates $\pi_t \leftarrow \Gamma_1^\lambda(\mu_t)$ and $\mu_{t+1} \leftarrow \Gamma_2(\pi_t, \mu_t)$ converge to the regularized NE at a linear rate, assuming that the exact optimal policy can be computed for $\Gamma_1^\lambda$.

## 3 Natural Actor-Critic for MFGs with Function Approximation

In this section, we present an online natural actor-critic (NAC) algorithm with function approximation to learn the regularized NE of a mean-field game (Algorithm 1). Four major steps are involved: policy evaluation, gradient estimation, mean-field update, and policy update.

For the policy update step, policy gradient methods improve the policy parameter $\theta$ by ascending along the direction of the gradient of the policy, i.e., $\nabla_\theta V_\mu^{\pi_\theta,\lambda}(\rho)$. A direct extension of the policy gradient theorem [68] shows that the policy gradient with entropy-regularization can be expressed as

$$\nabla_\theta V_\mu^{\pi_\theta,\lambda}(\rho) = \frac{1}{1-\gamma}\mathbb{E}_{s \sim d_\mu^{\pi_\theta}, a \sim \pi_\theta(\cdot|s)}\left[\nabla_\theta \log \pi_\theta(a|s) q_\mu^{\pi_\theta,\lambda}(s,a)\right].$$

Compared with the "vanilla" policy gradient method that follows the steepest direction in the parameter space, the natural policy gradient (NPG) approach [35] proceeds along the steepest direction with respect to the Fisher metric. NPG has the advantage of being invariant to the parameterization of the policy [4] and enjoys faster convergence as it follows a more direct path to the optimal solution. For a policy $\pi_\theta$ parameterized by $\theta$, NPG [35] defines a Fisher information matrix under policy $\pi_\theta$ as: $F^\theta = \mathbb{E}_{s \sim d_\mu^{\pi_\theta}, a \sim \pi_\theta(\cdot|s)}\left[\nabla_\theta \log \pi_\theta(a|s)\left(\nabla_\theta \log \pi_\theta(a|s)\right)^\top\right]$. NPG then performs gradient updates along the steepest direction induced by this matrix: $\theta \leftarrow \theta + \eta(F^\theta)^\dagger\nabla_\theta V_\mu^{\pi,\lambda}(\rho)$, where $\eta$ is the learning rate, and $(F^\theta)^\dagger$ denotes the Moore-Penrose pseudoinverse of $F^\theta$. Leveraging the notion of compatible function approximation, it can be shown that the above update rule can be equivalently expressed as (see [35] for a proof of the unregularized case and [13] for the regularized counterpart) $\theta \leftarrow \theta + \frac{\eta}{1-\gamma}w_\lambda^\theta$, where $w_\lambda^\theta$ is a minimizer of the following regression problem:

$$w_\lambda^\theta \in \operatorname*{argmin}_{w \in \mathbb{R}^d} \bar{L}(w, \theta), \text{ where } \bar{L}(w, \theta) \overset{\text{def}}{=} \mathbb{E}_{s \sim d_\mu^{\pi_\theta}, a \sim \pi_\theta(\cdot|s)}\left[\left(w^\top\nabla_\theta \log \pi_\theta(a|s) - q_\mu^{\pi_\theta,\lambda}(s,a)\right)^2\right].$$

For variance reduction purposes, Algorithm 1 further subtracts a baseline from the Q-function and

solves a variant of the regression problem at the $t$-th iteration instead:

$$w_t \in \underset{w \in \mathbb{R}^d : \|w\|_2 \leq R}{\operatorname{argmin}} L(w, \theta_t), \text{ where } L(w, \theta) \overset{\text{def}}{=} \mathbb{E}_{s \sim d_\mu^{\pi_\theta}, a \sim \pi_\theta(\cdot|s)} \left[ \left( w^\top \nabla_\theta \log \pi_\theta(a|s) - A_\mu^{\pi_\theta, \lambda}(s, a) \right)^2 \right],$$

where $A_\mu^{\pi, \lambda}(s, a) \overset{\text{def}}{=} Q_\mu^{\pi, \lambda}(s, a) - \mathbb{E}_{a' \sim \pi(\cdot|s)}[Q_\mu^{\pi, \lambda}(s, a')]$, and $R > 0$ is the gradient clipping radius. Since the target function may not be perfectly represented by a linear function, we use the *function approximation error* $\varepsilon_{\text{approx}}$ to denote the minimal possible error for our parametric class[2]:

$$\varepsilon_{\text{approx}} \overset{\text{def}}{=} \sup_{t \geq 0} \min_{w \in \mathbb{R}^d : \|w\|_2 \leq R} L(w, \theta_t).$$

In practice, the values of $A_\mu^{\pi_\theta, \lambda}$ and $w_t$ in the above regression problem cannot be calculated precisely and need to be computed from a finite number of samples, which further introduce *statistical errors* (or excess risk). In particular, we use the policy evaluation step (Line 4 of Algorithm 1) to compute an approximation $\hat{q}_t^\lambda$ of the shifted Q-function $q_{\mu_t}^{\pi_t, \lambda}$, where $\pi_t \overset{\text{def}}{=} \pi_{\theta_t}$ denotes the policy at the $t$-th iteration of Algorithm 1, and we apply the gradient estimation step (Line 5) to further use $\hat{q}_t^\lambda$ to calculate a gradient estimate $\hat{w}_t$. For ease of presentation, in this section, we will assume that we can obtain the estimated values $\hat{q}_t^\lambda$ and $\hat{w}_t$ from two black-box oracles. In Appendix C, we show that such oracles can be accomplished by standard sample-based estimation techniques. Specifically, we assume for now the existence of a policy evaluation oracle that returns an estimate $\hat{q}_t^\lambda$ such that

$$\mathbb{E}_{s \sim d_{\mu_t}^{\pi_t}, a \sim \pi_t(\cdot|s)} \left[ \left( \hat{q}_t^\lambda(s, a) - q_{\mu_t}^{\pi_t, \lambda}(s, a) \right)^2 \right] \leq \varepsilon_{\text{critic}}, \tag{2}$$

for some critic error $\varepsilon_{\text{critic}} \geq 0$. Let $\hat{Q}_t^\lambda(s, a) = \hat{q}_t^\lambda(s, a) + \lambda \log \pi_t(a|s)$, and $\hat{A}_t^\lambda(s, a) = \hat{Q}_t^\lambda(s, a) - \mathbb{E}_{a \sim \pi_t(\cdot|s)}[\hat{Q}_t^\lambda(s, a')]$. We further assume that a gradient estimation oracle provides an estimate $\hat{w}_t$ that satisfies:

$$\mathbb{E}\left[ (\hat{w}_t^\top \nabla \log \pi_t(a|s) - \hat{A}_t^\lambda(s, a))^2 \right] - \min_w \mathbb{E}\left[ (w^\top \nabla \log \pi_t(a|s) - \hat{A}_t^\lambda(s, a))^2 \right] \leq \varepsilon_{\text{actor}}, \tag{3}$$

for some actor error $\varepsilon_{\text{actor}} \geq 0$, where the expectation is over $s \sim d_{\mu_t}^{\pi_t}$ and $a \sim \pi_t(\cdot|s)$. $\varepsilon_{\text{critic}}$ and $\varepsilon_{\text{actor}}$ together represent the statistical error that can be driven to 0 when we have more samples, while $\varepsilon_{\text{approx}}$ accounts for the inherent modeling error due to the potential lack of expressiveness of the parametric policy class. We further use $\varepsilon_{\text{total}} = \varepsilon_{\text{approx}} + \varepsilon_{\text{actor}} + \varepsilon_{\text{critic}}$ to sum up all sources of errors.

A final remark on the policy update step is that we adopt the idea of gradient averaging from [13] to encourage exploration. Specifically, the parameter update in Line 7 is effectively $\theta_{t+1} \leftarrow (1 - \eta_t \lambda)\theta_t + \eta_t \lambda \cdot \frac{\hat{w}_t}{\lambda}$, which can be viewed as a convex combination of $\theta_t$ and $\hat{w}_t/\lambda$. Since our gradient estimation step "clips" the gradient estimate to ensure that $\|\hat{w}_t\|_2 \leq R$ for some $R > 0$ (Line 5), we can show by induction that the policy parameter is uniformly bounded, i.e., $\|\theta_t\|_2 \leq R/\lambda, \forall t \geq 0$. Together with the softmax parameterization of the policy, this condition ensures that our policy always explores the action space with some positive probability; that is, there exists $p_{\min} > 0$, such that $\pi_t(a|s) \geq p_{\min}, \forall t \geq 0, (s, a) \in \mathcal{S} \times \mathcal{A}$. Such a property is essential in establishing the convergence of the policy. See Lemma 2 in Appendix A for a formal treatment.

In each iteration of the algorithm, we update the mean-field state (Line 6) as $\mu_{t+1} \leftarrow (1 - \beta_t)\mu_t + \beta_t \Gamma_2(\pi_t, \mu_t)$, which can be considered as a "soft" step of mean-field evolution along the direction of $\Gamma_2(\pi_t, \mu_t)$ with a step size $\beta_t$. Similar to existing works [66, 28, 73, 78], this step assumes access to a simulator that returns the new population distribution given the current mean-field and policy. In practice, one can approximate the simulator by estimating the new population distribution through randomly sampling the actual states of a large number $N$ of agents, which incurs an estimation error $O(1/\sqrt{N})$ that decays as more agents are sampled [78].

Following [78], our algorithm enjoys the additional advantage of being *online* (also termed "learning while playing" in [78]), in the sense that it lets the mean-field simultaneously evolve as the agents play. Specifically, we perform only a single step of policy update for each iteration of mean-field evolution, instead of computing the exact optimal policy with respect to the current mean-field. This is achieved by carefully tuning the step sizes of the policy update ($\eta_t \approx T^{-2/5}$) and mean-field

---

[2]It has been shown that the approximation error is 0 for the realizable cases such as tabular (finite state-action) MDPs or linear MDPs [31], where the value functions are linear in the given features.

update $(\beta_t \approx T^{-4/5})$ so as to let the policy converge at a faster timescale than the mean-field does. Our online approach is in sharp contrast to the "double-loop" methods [23, 28, 5, 20, 49, 27] that have to fix the mean-field state as an outer loop and use an iterative process as the inner loop to learn an optimal policy with respect to each fixed population distribution, which can be more time consuming and incurs oscillations in the learned policies (as evidenced by our simulations).

## 4   Convergence Analysis

In this section, we theoretically establish the convergence of the natural actor-critic algorithm with linear function approximation to the regularized Nash equilibrium $(\pi^\star, \mu^\star)$ of the mean-field game. Our analysis follows the analytical framework of [78] for the tabular case. We start by introducing a few notations. For any $t \geq 0$, we use $\pi_t^\star$ to denote the optimal policy of the representative agent with respect to the mean-field state $\mu_t$, i.e., $\pi_t^\star = \Gamma_1^\lambda(\mu_t)$. Let $d_t^\star \overset{\text{def}}{=} d_{\mu_t}^{\pi_t^\star}$ be the visitation distribution of $\pi_t^\star$ under $\mu_t$, and $d^\star \overset{\text{def}}{=} d_{\mu^\star}^{\pi^\star}$ be the one induced by the NE $(\pi^\star, \mu^\star)$. Our first result establishes the improvement of the policy under natural actor-critic updates in terms of the KL divergence. All missing proofs are deferred to Appendix B.

**Lemma 1.** *(Policy improvement). For any time $t \geq 0$, the policy update $\theta_{t+1} = \theta_t + \eta_t g_t$ with softmax parameterization and linear function approximation leads to the following policy improvement:*

$$\mathbb{E}_{s \sim d_t^\star}\left[KL(\pi_t^\star(\cdot|s) \| \pi_{t+1}(\cdot|s))\right] \leq (1 - \eta_t \lambda)\mathbb{E}_{s \sim d_t^\star}\left[KL(\pi_t^\star(\cdot|s)) \| \pi_t(\cdot|s))\right] - \eta_t \mathbb{E}_{s \sim d_t^\star}[V_{\mu_t}^{\pi_t, \lambda}(s)]$$

$$- \eta_t \mathbb{E}_{s \sim d_t^\star, a \sim \pi_t^\star(\cdot|s)}\left[g_t^\top \nabla_\theta \log \pi_t(a|s) - q_\mu^{\pi_t, \lambda}(s, a)\right] + \frac{1}{2}\eta_t^2 \|g_t\|_2^2.$$

The proof follows from the performance difference lemma [34] in the regularized case and the smoothness of $\log \pi_\theta(a|s)$. Similar results have also been shown in [13, 78]. To proceed further, we impose the following regularity assumptions on the visitation distribution.

**Assumption 1.** *There exists a constant $d_0 > 0$, such that for any mean-field states $\mu$ and $\mu'$, the state visitation distributions under their corresponding optimal policies $\pi^\star$ and $\pi^{\star\prime}$ satisfy $\|d_\mu^{\pi^\star} - d_{\mu'}^{\pi^{\star\prime}}\|_1 \leq d_0 \|\mu - \mu'\|_1$.*

**Assumption 2.** *(Finite concentrability coefficients). There exist constants $C_1, C_2, C_3 > 0$, such that for any $t \geq 0$ and any encountered mean-field state $\mu_t$,*

$$\sup_{s \in \mathcal{S}} \frac{d_t^\star(s)}{d^\star(s)} \leq C_1, \quad \mathbb{E}_{s \sim d_t^\star}\left[\left|\frac{d^\star(s)}{d_t^\star(s)}\right|^2\right] \leq C_2^2, \quad \text{and} \quad \mathbb{E}_{s \sim d_{\mu_t}^{\pi_t}}\left[\left|\frac{d_t^\star(s)}{d_{\mu_t}^{\pi_t}(s)}\right|^2\right] \leq C_3^2.$$

Assumption 1 states that the visitation distributions are smooth w.r.t the mean fields, which is consistent with [78] and is reminiscent of the smooth visitation assumption for RL in non-stationary environments [21]. Assumption 2 is a standard assumption in policy optimization [34, 63, 2, 78, 13] that captures the difficulty of strategic exploration by requiring the visitation distributions of certain policies to adequately cover that of an optimal policy. Together with Lemma 1, these assumptions allow us to establish a recursive relationship of $\text{KL}(\pi_t^\star \| \pi_t)$ over time (Lemma 8 in Appendix B).

To characterize the convergence of $\pi_t$, we use $D(\pi, \pi') \overset{\text{def}}{=} \mathbb{E}_{s \sim d^\star}[\|\pi(\cdot|s) - \pi'(\cdot|s)\|_1]$ as a measure of distance between two policies. The following assumption is standard in the literature [28, 73, 78, 16] that imposes the Lipschitzness of the two operators $\Gamma_1^\lambda$ and $\Gamma_2$ with respect to the $D(\cdot, \cdot)$ metric.

**Assumption 3.** *(Lipschitz Operators). There exist constants $d_1, d_2, d_3 > 0$, such that for any policies $\pi, \pi'$ and mean-field states $\mu, \mu'$, it holds that: (1) $D\left(\Gamma_1^\lambda(\mu), \Gamma_1^\lambda(\mu')\right) \leq d_1 \|\mu - \mu'\|_1$, $\|\Gamma_2(\pi, \mu) - \Gamma_2(\pi', \mu)\|_1 \leq d_2 D(\pi, \pi')$, and (2) $\|\Gamma_2(\pi, \mu) - \Gamma_2(\pi, \mu')\|_1 \leq d_3 \|\mu - \mu'\|_1$.*

The first condition states that $\Gamma_1^\lambda(\mu)$ is Lipschitz with respect to the mean-field state, and the second and third conditions stipulate that $\Gamma_2(\pi, \mu)$ is Lipschitz in each of its arguments when fixing the other. Assumption 3 immediately implies that the composite operator $\Lambda^\lambda$ is contractive when the Lipschitz constants are small enough (Lemma 6 in Appendix A). We are now ready to state our main theoretical guarantees on the convergence of the policy-population sequence $\{(\pi_t, \mu_t)\}_{t \geq 0}$ to the NE $(\pi^\star, \mu^\star)$.

**Theorem 1.** *Suppose that Assumptions 1 − 3 hold with $\bar{d} = 1 - d_1 d_2 - d_3 > 0$, and the policy evaluation and gradient estimation oracles satisfy (2) and (3), respectively. The policy-population*

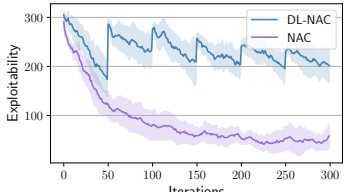
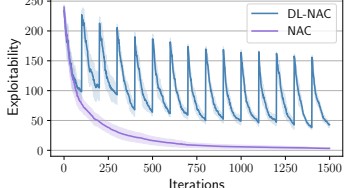
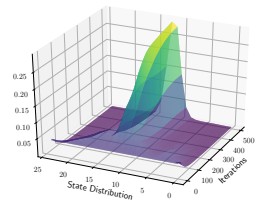

Figure 1: SIS Exploitability      Figure 2: LQ Exploitability      Figure 3: LQ Evolution

*sequence $\{(\pi_t, \mu_t)\}_{t \geq 0}$ generated by Algorithm 1 satisfies*

$$D\left(\pi^\star, \frac{1}{T}\sum_{t=0}^{T-1}\pi_t\right) + \left\|\mu^\star - \frac{1}{T}\sum_{t=0}^{T-1}\mu_t\right\|_1 \leq \widetilde{O}\left(\frac{1}{\lambda T^{1/5}} + \sqrt{\frac{|\mathcal{A}|\exp(1/\lambda)\varepsilon_{total}^{1/2}}{\lambda}}\right)$$

Theorem 1 establishes the finite-time convergence of Algorithm 1. After $T$ iterations, the averaged policy-population pair $(\frac{1}{T}\sum_{t=0}^{T-1}\pi_t, \frac{1}{T}\sum_{t=0}^{T-1}\mu_t)$ constitutes an $\widetilde{O}(T^{-1/5})$-approximate NE, subject to the approximation and statistical errors. As we collect more samples, the statistical errors can be driven to 0. However, the approximation error $\varepsilon_{\text{approx}}$ is an inherent mis-modeling cost due to using a (potentially not expressive enough) linear function to approximate the policy, which might not diminish to 0 regardless of the number of iterations we run. While this seems to imply that linear function approximation can be restrictive, our experimental results (Section 5) instead suggest that linear functions are already satisfactory in most application scenarios that we are interested in. The convergence rate $\widetilde{O}$ remains the same as the tabular case [78], indicating that linear function approximation does not introduce intractability in terms of the dependence on $T$. The convergence rate in Theorem 1 is independent of the size of the state space $|\mathcal{S}|$, implying that our approach is applicable to MFGs with large state spaces. The dependence on the action space $|\mathcal{A}|$ is also mild, and in mostly RL-related tasks the action space is adequately small [44]. The convergence rate is also inverse proportional to the regularization parameter $\lambda$. This indicates that a higher regularization level can accelerate the convergence of the algorithm while introducing a larger regularization error according to (1) on the other hand. In practice, one can choose a regularization level that balances the convergence rate and accuracy.

## 5 Experimental Results

### 5.1 Simulations on Classic MFGs

We first evaluate our natural actor-critic algorithm on two classic mean-field games considered in the literature, including an SIS epidemics model [16, 38], and a linear-quadratic MFG [49, 38, 12, 45]. We refer to these two tasks as SIS and LQ, respectively. Detailed descriptions of the tasks and the simulation setups are deferred to Appendix D. We implement Algorithm 1 with linear function approximation ("NAC" for short) and use temporal difference (TD) learning also with linear function approximation as the critic for policy evaluation. We utilize the standard notion of *exploitability* [84, 49, 16] to measure the sub-optimality of a policy. Intuitively, a higher degree of exploitability suggests that an individual agent can be much better off by deviating from the given policy. As a comparison baseline, we also implement a "double-loop" version of natural actor-critic ("DL-NAC") that uses a fixed point iteration in a similar fashion as existing works [28, 5, 27].

Simulation results on SIS and LQ are given in Figures 1 and 2, respectively. All results are averaged over 10 runs. For both tasks, we can observe that the exploitability of NAC converges to 0, indicating that NAC can efficiently learn the NE. DL-NAC also converges in general, but it suffers "zigzag" fluctuations due to the fact that double-loop methods update the mean-field abruptly and hence nullify the policies learned from the past. See Appendix D for a detailed discussion of this phenomenon. Similar patterns have also been observed in the literature [16]. Hence, our online method enjoys faster convergence and more smooth learning behavior than the fixed-point iteration. In Figure 3, we further plot the evolution of the state distribution over time when applying NAC to LQ. The population starts from a uniform distribution over the state space, and as time goes by, it quickly concentrates in a small neighborhood of the state space, a desired behavior for linear-quadratic tracking-type problems [72].

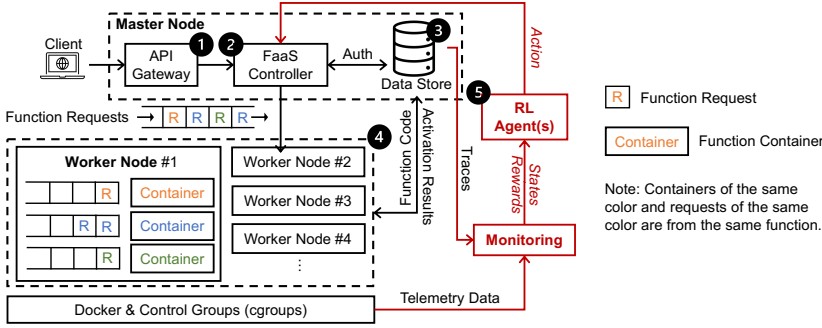

Figure 4: OpenWhisk cluster architecture and function container resource management with reinforcement learning. Each RL agent is assigned to one function.

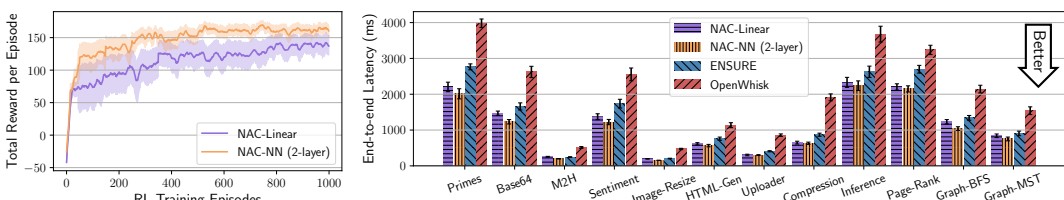

Figure 5: Policy Training

Figure 6: Policy Serving.

## 5.2 Experiments on a Serverless Platform

We apply our approach (with both linear and neural-network function approximations) to a resource management problem in a serverless computing environment. Here, we directly consider a finite-agent game and collect the empirical average of the local states as the mean-field.

**Serverless Platform and Workload.** We use a production-grade open-source serverless platform, OpenWhisk [22], and deploy it on IBM Cloud with 22 VMs. The OpenWhisk cluster consists of one master node (that runs the serverless controller) and 21 worker nodes (each of which runs the function containers). Figure 4 shows the architecture of a distributed OpenWhisk cluster and how RL agents work with the OpenWhisk cluster to manage resources of each function. The master node runs the API gateway (labeled as ❶), FaaS controller (❷), data store (❸), and other management modules. Each of the worker nodes (labeled as ❹) runs the function containers. We deploy the workload generator and RL agents from two separate nodes in the same cluster and use FaaSProfiler [60] to trace requests for function latency measurements. We select diverse function benchmarks from widely used open-source FaaS benchmark suites [15, 60, 83], including web applications (e.g., `HTML-Gen`), multimedia application (e.g., `Image-Resize`), scientific functions (e.g., `PageRank`), and ML-model serving (e.g., `Sentiment-Analysis`). To drive the benchmarks, we sample and replay the function invocations from Azure function traces [61]. Unlike [51], however, we do not explicitly consider function dependencies, but we believe that our approach is orthogonal to [51] and can be potentially integrated with the critical component localization of [51] to model function dependencies. More detailed descriptions of the OpenWhisk setup are relegated to Appendix E.

**RL Pipeline in OpenWhisk.** We model the resource management for each serverless function as a sequential decision-making problem that can be captured by RL. At each step in the sequence, an RL agent (labeled as ❺ in Figure 4) monitors the system and application conditions from both the OpenWhisk data store and the Linux cgroups. The collected measurements include function-level performance statistics (i.e., tail latencies on execution time, waiting time, and cold-start time for serving function requests) and system-level resource utilization statistics (e.g., CPU utilization of function containers). These measured telemetry data are pre-processed and used to define RL states and rewards, which is then mapped to a resource management decision by the RL agent. The decision is finally executed by FaaS controller and takes effect on the OpenWhisk platform. After applying the action, the RL agent receives another state and reward in the next time step.

**MARL Formulation.** We formulate the function resource management problem on a multi-tenant serverless platform as a Markov game where each function is assigned to one agent. At each step, the agent perceives available system and application conditions (e.g., resource utilization and function latencies) from the platform monitor as the state. The action space includes both horizontal scaling (i.e., scaling the number of function containers) and vertical scaling (i.e., scaling the container size). Since the objective is to meet QoS objectives while keeping the resource utilization at a high level, we consider a reward function as $r_t = \alpha \cdot QP(t) + (1-\alpha)/2 \cdot (RU_{cpu}(t) + RU_{mem}(t)) + penalty$, where $QP(t)$ and $RU(t)$ are the QoS preservation ratio and resource utilization at time $t$, and $penalty$ is set to -1 (and 0 otherwise) for illegal or undesired actions (e.g., dangling decisions). We implement and evaluate two variants of our method: one exploits linear function approximation for both the actor and the critic ("NAC-Linear" for short), and the other one leverages a two-layer fully-connected neural network as function approximation (i.e., "NAC-NN"). Since serverless functions usually have diverse resource requirements and behaviors, we resort to a multi-type formulation of MFGs (see [48, 75, 24] for more extensive discussions) to allow for heterogeneous agents.

**Policy Training.** For training, we create 50 functions[3] on OpenWhisk, each of which is randomly selected from the function benchmarks. Figure 5 shows the total reward achieved by NAC-Linear and NAC-NN at each training iteration. We observe that NAC-NN takes about 300 fewer episodes to converge and achieves an 18.8% higher reward after convergence compared to NAC-Linear. We attribute this to the fact that linear function approximation cannot fully capture the complex system dynamics and decision-making policies compared to neural networks.

**Policy Serving.** We measure the policy-serving performance using two metrics: the 99th percentile function latency (commonly used in user-defined QoS objectives) and the number of function containers in use. We take the model checkpoints of NAC agents after convergence and leverage the learned policies to manage resources for each function. Figure 6 shows the function performance managed by NAC-Linear and NAC-NN compared with two heuristics-based approaches ENSURE [67] and Open-Whisk's original resource manager (Appendix E). As shown in Figure 6, NAC-NN achieves the best performance and has 33.6% to 67.3% (for `Image-Resize`) lower latency compared to OpenWhisk's original algorithm. NAC-NN also has 14.8% to 29.6% (for `Sentiment-Analysis`) lower latency compared to ENSURE. Additionally, NAC-NN uses 29% fewer function containers than ENSURE and 37% fewer than OpenWhisk's design, indicating a higher resource utilization level. NAC-Linear also outperforms ENSURE regarding function tail latency (by up to 25.5%, for `Uploader`) and resource utilization (by up to 24% fewer function containers overall), which suggests that the simple linear function approximation can already lead to reasonable performances in practice.

## 6 Concluding Remarks

In this paper, we have proposed a mean-field game approach to large-scale cloud resource management. We have presented an online natural actor-critic algorithm and proved its finite-time convergence to the regularized Nash equilibrium with linear function approximation. We have evaluated our solution on a serverless platform using both linear and neural-network function approximations, which has demonstrated superior performances in terms of scalability, latency, and resource utilization. Interesting future directions include further tightening the convergence rate and investigating alternative forms of function approximations for other application scenarios.

## Acknowledgments and Disclosure of Funding

This work was partially supported by the National Science Foundation (NSF) under grant CCF 20-29049; by the IBM-ILLINOIS Center for Cognitive Computing Systems Research (C3SR), a research collaboration that is part of the IBM AI Horizon Network; by the IBM-ILLINOIS Discovery Accelerator Institute (IIDAI); and by the Air Force Office of Scientific Research (AFOSR) under grant FA9550-19-1-0353. Any opinions, findings or recommendations expressed in this material are those of the authors and do not necessarily reflect the views of the NSF or IBM or AFOSR.

---

[3]We do not experiment with even more functions because the serverless platform architecture is a centralized model where the central manager usually becomes the bottleneck for scalability. To scale beyond thousands of functions in the production environment, cloud datacenters can use a two-tier model where a large cluster is divided into a couple of sub-clusters [17]. In this case, our method can be applied to each system sub-cluster to address the scalability issue.

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
