# Supplementary Materials for "A Mean-Field Game Approach to Cloud Resource Management with Function Approximation"

## A    Technical Lemmas

**Lemma 2.** *(Persistence of excitation, Lemma 2 of [13]). For any $\lambda > 0$, the entropy-regularized natural actor-critic update with averaging satisfies*

$$\|\theta_t\|_2 \leq R/\lambda, \forall t \geq 0, \quad and \quad p_{\min} = \inf_{t \geq 0} \min_{(s,a) \in \mathcal{S} \times \mathcal{A}} \pi_t(a|s) \geq \frac{\exp(-2R/\lambda)}{|\mathcal{A}|} > 0.$$

*Proof.* We prove the first statement by induction. The statement holds for $t = 0$ due to our initialization $\theta_0 = \mathbf{0}$. Expanding the recursive policy update rule

$$\theta_{t+1} = (1 - \eta_t \lambda)\theta_t + \eta_t \hat{w}_t = (1 - \eta_t \lambda)\theta_t + \eta_t \lambda \cdot \frac{\hat{w}_t}{\lambda}.$$

Applying the triangle inequality,

$$\|\theta_{t+1}\|_2 \leq (1 - \eta_t \lambda) \|\theta_t\|_2 + \eta_t \lambda \|\hat{w}_t/\lambda\|_2 \leq R/\lambda,$$

where the last step holds because of the induction hypothesis, the fact that our gradient estimation step guarantees $\|\hat{w}_t\|_2 \leq R$, and that $0 < \eta_t \lambda < 1$. Invoking the induction completes the proof of $\|\theta_t\|_2 \leq R/\lambda, \forall t \geq 0$.

Further, using the condition that $\|\phi_{s,a}\|_2 \leq 1$, the Cauchy-Schwarz inequality implies that $|\theta_t^\top \phi_{s,a}| \leq R/\lambda, \forall (s,a) \in \mathcal{S} \times \mathcal{A}$. Under softmax parameterization,

$$p_{\min} = \inf_{t \geq 0} \min_{(s,a) \in \mathcal{S} \times \mathcal{A}} \pi_t(a|s) \geq \frac{\exp(-R/\lambda)}{|\mathcal{A}| \exp(R/\lambda)} = \frac{\exp(-2R/\lambda)}{|\mathcal{A}|} > 0.$$

This completes the proof of the lemma.    □

**Lemma 3.** *Let $Q_{\max} = \frac{1 + \gamma \lambda \log |\mathcal{A}|}{1 - \gamma}$. For any mean-field state $\mu$, the optimal policy $\pi_\mu^{\star,\lambda}$ with respect to the MDP induced by $\mu$ satisfies that*

$$\pi_\mu^{\star,\lambda}(a|s) \geq \frac{1}{|\mathcal{A}| \exp(Q_{\max}/\lambda)}, \forall (s,a) \in \mathcal{S} \times \mathcal{A}.$$

*Proof.* It has been shown [46] that the optimal policy $\pi_\mu^{\star,\lambda}$ can be expressed as a Boltzmann distribution of the form

$$\pi_\mu^{\star,\lambda}(a|s) \propto \exp\left(\frac{Q_\mu^{\star,\lambda}(s,a)}{\lambda}\right),$$

where $Q_\mu^{\star,\lambda}(s,a)$ is the optimal soft Q-function. From the definition of $Q_\mu^{\pi,\lambda}$ and the facts that $r(s,a,\mu) \in [0,1]$ and $\mathcal{H}(p) \leq \log |\mathcal{A}|$ for any distribution $p$ over $\mathcal{A}$, we can easily see that $Q_\mu^{\star,\lambda}(s,a) \leq Q_{\max} = \frac{1 + \gamma \lambda \log |\mathcal{A}|}{1 - \gamma}$. Therefore, for any $(s,a) \in \mathcal{S} \times \mathcal{A}$,

$$\pi_\mu^{\star,\lambda}(a|s) = \frac{\exp\left(Q_\mu^{\star,\lambda}(s,a)/\lambda\right)}{\sum_{b \in \mathcal{A}} \exp\left(Q_\mu^{\star,\lambda}(s,b)/\lambda\right)} \geq \frac{1}{\sum_{b \in \mathcal{A}} \exp\left(Q_{\max}/\lambda\right)} = \frac{1}{|\mathcal{A}| \exp(Q_{\max}/\lambda)}.$$

□

**Lemma 4.** *(Lemma 3 of [78]). For any $x, y, z \in \mathcal{A}$, if $x(a) \geq \alpha_1, y(a) \geq \alpha_1$, and $z(a) \geq \alpha_2, \forall a \in \mathcal{A}$, then*

$$KL(x\|z) - KL(y\|z) \leq \left(1 + \log \frac{1}{\min\{\alpha_1, \alpha_2\}}\right) \cdot \|x - y\|_1.$$

**Lemma 5.** *Under the same conditions as Lemma 8, it holds that*

$$\sigma_{t+1}^{\pi} \le \mathbb{E}_{d_t^{\star}}\left[KL(\pi_t^{\star}\|\pi_{t+1})\right] + (1 + C_1) \cdot \kappa d_0 \left\|\mu_{t+1} - \mu_t\right\|_1,$$

*where* $\kappa = \frac{2\log|\mathcal{A}|}{1-\gamma} + \frac{1+2R(1-\gamma)}{\lambda(1-\gamma)}$.

*Proof.* From the definition of $\sigma_{t+1}^{\pi}$,

$$\sigma_{t+1}^{\pi} = \mathbb{E}_{d_{t+1}^{\star}}\left[\mathrm{KL}(\pi_{t+1}^{\star}\|\pi_{t+1})\right]$$

$$\le \mathbb{E}_{d_{t+1}^{\star}}\left[\mathrm{KL}(\pi_t^{\star}\|\pi_{t+1})\right] + \left|\mathbb{E}_{d_{t+1}^{\star}}\left[\mathrm{KL}(\pi_{t+1}^{\star}\|\pi_{t+1}) - \mathrm{KL}(\pi_t^{\star}\|\pi_{t+1})\right]\right|$$

$$= \mathbb{E}_{d_t^{\star}}\left[\mathrm{KL}(\pi_t^{\star}\|\pi_{t+1})\right] + (\mathbb{E}_{d_{t+1}^{\star}} - \mathbb{E}_{d_t^{\star}})\left[\mathrm{KL}(\pi_t^{\star}\|\pi_{t+1})\right] + \left|\mathbb{E}_{d_{t+1}^{\star}}\left[\mathrm{KL}(\pi_{t+1}^{\star}\|\pi_{t+1}) - \mathrm{KL}(\pi_t^{\star}\|\pi_{t+1})\right]\right| \tag{4}$$

In the following, we upper bound each term in (4) separately. We first define $p_{\min} \overset{\text{def}}{=} \inf_{t\ge 0}\min_{(s,a)\in\mathcal{S}\times\mathcal{A}}\pi_t(a|s)$, and apply the persistence of excitation condition from Lemma 2 to obtain that $p_{\min} \ge \frac{\exp(-2R/\lambda)}{|\mathcal{A}|} > 0$. To upper bound the second term in (4), we first show that for any $s \in \mathcal{S}$,

$$\mathrm{KL}(\pi_t^{\star}(\cdot|s)\|\pi_{t+1}(\cdot|s)) = \sum_{a\in\mathcal{A}}\pi_t^{\star}(a|s)\log\frac{\pi_t^{\star}(a|s)}{\pi_{t+1}(a|s)} \le \sum_{a\in\mathcal{A}}\pi_t^{\star}(a|s)\log\frac{1}{p_{\min}} \le \log|\mathcal{A}| + 2R/\lambda.$$

If we define $\mathrm{KL}_{\max} \overset{\text{def}}{=} \log|\mathcal{A}| + 2R/\lambda$, we will have that

$$(\mathbb{E}_{d_{t+1}^{\star}} - \mathbb{E}_{d_t^{\star}})\left[\mathrm{KL}(\pi_t^{\star}\|\pi_{t+1})\right] = \mathbb{E}_{s\sim d^{\star}}\left[\frac{d_{t+1}^{\star}(s) - d_t^{\star}(s)}{d^{\star}(s)} \cdot \mathrm{KL}(\pi_t^{\star}\|\pi_{t+1})\right]$$

$$\le \mathrm{KL}_{\max} \cdot \mathbb{E}_{s\sim d^{\star}}\left[\frac{|d_{t+1}^{\star}(s) - d_t^{\star}(s)|}{d^{\star}(s)}\right]$$

$$\le \mathrm{KL}_{\max} \cdot d_0\left\|\mu_{t+1} - \mu_t\right\|_1, \tag{5}$$

where the last step follows from Assumption 1. This gives an upper bound of the second term.

We proceed to upper bound the third term in (4). Let $\tau = \frac{1}{|\mathcal{A}|}\exp\left(-\frac{1+\gamma\lambda\log|\mathcal{A}|}{\lambda(1-\gamma)}\right)$. From Lemma 3, we know that

$$\pi_t^{\star}(a|s) \ge \tau, \text{ and } \pi_{t+1}^{\star}(a|s) \ge \tau, \forall (s,a) \in \mathcal{S}\times\mathcal{A}.$$

Since both $\pi_t(a|s)$ and $\pi_t^{\star}(a|s)$ are lower bounded, we can apply the Lipschitzness of KL-divergence (Lemma 4) and obtain

$$\left|\mathbb{E}_{d_{t+1}^{\star}}\left[\mathrm{KL}(\pi_{t+1}^{\star}\|\pi_{t+1}) - \mathrm{KL}(\pi_t^{\star}\|\pi_{t+1})\right]\right|$$

$$\le \left(1 + \log\frac{1}{\min\{\tau, p_{\min}\}}\right)\mathbb{E}_{s\sim d_{t+1}^{\star}}\left[\left\|\pi_t^{\star}(\cdot|s) - \pi_{t+1}^{\star}(\cdot|s)\right\|_1\right]$$

$$\le \kappa\mathbb{E}_{s\sim d^{\star}}\left[\frac{d_{t+1}^{\star}(s)}{d^{\star}(s)} \cdot \left\|\pi_t^{\star}(\cdot|s) - \pi_{t+1}^{\star}(\cdot|s)\right\|_1\right]$$

$$\le \kappa C_1 D(\pi_t^{\star}, \pi_{t+1}^{\star})$$

$$= \kappa C_1 D(\Gamma_1^{\lambda}(\mu_t), \Gamma_1^{\lambda}(\mu_{t+1}))$$

$$\le \kappa C_1 d_0\left\|\mu_t - \mu_{t+1}\right\|_1, \tag{6}$$

where the third inequality is by Assumption 2, the last step is due to Assumption 1, and

$$\kappa \overset{\text{def}}{=} \frac{2\log|\mathcal{A}|}{1-\gamma} + \frac{1+2R(1-\gamma)}{\lambda(1-\gamma)}$$

$$\ge 1 + \max\left\{\log|\mathcal{A}| + \frac{1+\gamma\lambda\log|\mathcal{A}|}{\lambda(1-\gamma)}, \log|\mathcal{A}| + 2R/\lambda\right\}$$

$$\ge 1 + \log\frac{1}{\min\{\tau, p_{\min}\}}.$$

This gives an upper bound of the third term in (4). Combining (4), (5), and (6) completes the proof of the lemma. $\qquad\square$

**Lemma 6.** *Under Assumption 3, it holds for any mean-field states $\mu, \mu'$ that*

$$\left\|\Lambda^\lambda(\mu) - \Lambda^\lambda(\mu')\right\|_1 \leq (d_1 d_2 + d_3) \left\|\mu - \mu'\right\|_1.$$

*In particular, $\Lambda^\lambda$ is a contraction if $d_1 d_2 + d_3 < 1$.*

*Proof.* By the definition of the composite operator,

$$
\begin{aligned}
\left\|\Lambda^\lambda(\mu) - \Lambda^\lambda(\mu')\right\|_1 &= \left\|\Gamma_2\left(\Gamma_1^\lambda(\mu), \mu\right) - \Gamma_2\left(\Gamma_1^\lambda(\mu'), \mu'\right)\right\|_1 \\
&\leq \left\|\Gamma_2\left(\Gamma_1^\lambda(\mu), \mu\right) - \Gamma_2\left(\Gamma_1^\lambda(\mu'), \mu\right)\right\|_1 + \left\|\Gamma_2\left(\Gamma_1^\lambda(\mu'), \mu\right) - \Gamma_2\left(\Gamma_1^\lambda(\mu'), \mu'\right)\right\|_1 \\
&\leq d_2 D\left(\Gamma_1^\lambda(\mu), \Gamma_1^\lambda(\mu')\right) + d_3 \left\|\mu - \mu'\right\|_1 \\
&\leq (d_1 d_2 + d_3) \left\|\mu - \mu'\right\|_1,
\end{aligned}
$$

where the second inequality uses the Lipschitzness of $\Gamma_2$, and the last inequality is due to the Lipschitzness of $\Gamma_1^\lambda$. $\qquad\square$

**Lemma 7.** *(Lemma 8 of [78]). Suppose that Assumptions 2 and 3 hold with $\bar{d} = 1 - d_1 d_2 - d_3 > 0$, we then have*

$$\left\|\mu_{t+1} - \mu^\star\right\|_1 \leq (1 - \beta_t \bar{d}) \left\|\mu_t - \mu^\star\right\|_1 + d_2 C_2 \beta_t \sqrt{\sigma_t^\pi}, \forall t \geq 0.$$

*Proof.* For notational convenience, define $\sigma_t^\mu \stackrel{\text{def}}{=} \left\|\mu_t - \mu^\star\right\|_1$. Since Algorithm 1 updates the mean-field state $\mu_t$ in the same way as [78], our $\sigma_t^\mu$ also exhibits the same recursive behavior as characterized in Lemma 8 of [78], and we reproduce the proof here for completeness. Using the update rule of the mean-field state in Algorithm 1,

$$
\begin{aligned}
&\left\|\mu_{t+1} - \mu^\star\right\|_1 \\
&= \left\|(1 - \beta_t)\mu_t + \beta_t \Gamma_2(\pi_t, \mu_t) - \mu^\star\right\|_1 \\
&= \left\|(1 - \beta_t)(\mu_t - \mu^\star) + \beta_t \left(\Gamma_2\left(\Gamma_1^\lambda(\mu_t), \mu_t\right) - \mu^\star\right) - \beta_t \left(\Gamma_2\left(\Gamma_1^\lambda(\mu_t), \mu_t\right) - \Gamma_2(\pi_t, \mu_t)\right)\right\|_1 \\
&\leq (1 - \beta_t) \left\|\mu_t - \mu^\star\right\|_1 + \beta_t \left\|\Gamma_2(\Gamma_1^\lambda(\mu_t), \mu_t) - \Gamma_2(\Gamma_1^\lambda(\mu^\star), \mu^\star)\right\|_1 \\
&\quad + \beta_t \left\|\Gamma_2(\Gamma_1^\lambda(\mu_t), \mu_t) - \Gamma_2(\pi_t, \mu_t)\right\|_1, \\
&\leq (1 - \beta_t \bar{d}) \left\|\mu_t - \mu^\star\right\|_1 + \beta_t d_2 D(\pi_t^\star, \pi_t), \qquad\qquad\qquad\qquad\qquad\qquad\qquad (7)
\end{aligned}
$$

where the first inequality uses the fact that $\Gamma_2(\Gamma_1^\lambda(\mu^\star), \mu^\star) = \mu^\star$, the second inequality follows from the Lipschitzness of the operators $\Lambda^\lambda$ (Lemma 6) and $\Gamma_2$. To further upper bound the second term on the RHS of (7), we recall the definition that

$$
\begin{aligned}
D(\pi_t^\star, \pi_t) &= \mathbb{E}_{s \sim d^\star}\left[\left\|\pi_t^\star(\cdot|s) - \pi_t(\cdot|s)\right\|_1\right] \\
&= \mathbb{E}_{s \sim d_t^\star}\left[\frac{d^\star(s)}{d_t^\star(s)} \cdot \left\|\pi_t^\star(\cdot|s) - \pi_t(\cdot|s)\right\|_1\right] \\
&\leq \left(\mathbb{E}_{s \sim d_t^\star}\left[\left|\frac{d^\star(s)}{d_t^\star(s)}\right|^2\right] \cdot \mathbb{E}_{s \sim d_t^\star}\left[\left\|\pi_t^\star(\cdot|s) - \pi_t(\cdot|s)\right\|_1^2\right]\right)^{\frac{1}{2}} \\
&\leq C_2 \sqrt{\mathbb{E}_{s \sim d_t^\star}\left[\text{KL}(\pi_t^\star(\cdot|s)\|\pi_t(\cdot|s)\right]}, \qquad\qquad\qquad\qquad\qquad\qquad (8)
\end{aligned}
$$

where the last step follows from Assumption 2 and Pinsker's inequality. Plugging (8) back to (7) completes the proof. $\qquad\square$

# B    Proofs for Section 4

## B.1    Proof for Lemma 1

*Proof.* First, notice that for any fixed $(s, a) \in \mathcal{S} \times \mathcal{A}$, the log-linear policy $\log \pi_\theta(a|s)$ is a 1-smooth function in $\theta$:

$$\left\|\nabla_\theta \log \pi_\theta(a|s) - \nabla_\theta \log \pi_{\theta'}(a|s)\right\|_2 \leq \left\|\theta - \theta'\right\|_2, \forall \theta, \theta' \in \mathbb{R}^d.$$

A standard result for an $L$-smooth function $f$ on $\mathbb{R}^n$ is that (e.g., Lemma 3.4 of [11])

$$f(y) \geq f(x) + \nabla f(x)^\top (y - x) - \frac{L}{2} \|y - x\|_2^2, \forall x, y \in \mathbb{R}^n.$$

We hence obtain that

$$\mathbb{E}_{s \sim d_t^\star} \left[ \mathrm{KL}(\pi_t^\star(\cdot|s) \| \pi_{t+1}(\cdot|s)) \right] - \mathbb{E}_{s \sim d_t^\star} \left[ \mathrm{KL}(\pi_t^\star(\cdot|s) \| \pi_t(\cdot|s)) \right]$$

$$\leq \mathbb{E}_{s \sim d_t^\star} \left[ \sum_{a \in \mathcal{A}} \pi_t^\star(a|s) \left( \log \pi_t(a|s) - \log \pi_{t+1}(a|s) \right) \right]$$

$$\leq - \eta_t \mathbb{E}_{s \sim d_t^\star, a \sim \pi_t^\star(\cdot|s)} \left[ g_t^\top \nabla_\theta \log \pi_t(a|s) \right] + \frac{\eta_t^2}{2} \|g_t\|_2^2$$

$$= - \eta_t \mathbb{E}_{s \sim d_t^\star, a \sim \pi_t^\star(\cdot|s)} \left[ g_t^\top \nabla_\theta \log \pi_t(a|s) - q_\mu^{\pi_t,\lambda}(s,a) \right] - \eta_t \mathbb{E}_{d_t^\star \circ \pi_t^\star} \left[ q_\mu^{\pi_t,\lambda}(s,a) \right] + \frac{\eta_t^2}{2} \|g_t\|_2^2.$$

The performance difference lemma in the regularized case (e.g., Lemma 5 of [13]) implies that for any two policies $\pi$ and $\pi'$, we have that

$$V_\mu^{\pi,\lambda}(\rho) - V_\mu^{\pi',\lambda}(\rho) = \frac{1}{1-\gamma} \mathbb{E}_{s \sim d_\mu^\pi, a \sim \pi(\cdot|s)} \left[ q_\mu^{\pi',\lambda}(s,a) - V_\mu^{\pi',\lambda}(s) \right] - \frac{\lambda}{1-\gamma} \mathbb{E}_{s \sim d_\mu^\pi} \left[ \mathrm{KL}(\pi(\cdot|s) \| \pi'(\cdot|s)) \right].$$

By letting $\mu = \mu_t, \pi' = \pi_t$ and $\pi = \pi_t^\star$, we further have that

$$\mathbb{E}_{s \sim d_t^\star} \left[ \mathrm{KL}(\pi_t^\star(\cdot|s) \| \pi_{t+1}(\cdot|s)) \right] - \mathbb{E}_{s \sim d_t^\star} \left[ \mathrm{KL}(\pi_t^\star(\cdot|s) \| \pi_t(\cdot|s)) \right]$$

$$\leq - \eta_t \mathbb{E}_{s \sim d_t^\star, a \sim \pi_t^\star(\cdot|s)} \left[ g_t^\top \nabla_\theta \log \pi_t(a|s) - q_\mu^{\pi_t,\lambda}(s,a) \right] - \eta_t(1-\gamma) \left( V_{\mu_t}^{\pi_t^\star,\lambda}(\rho) - V_{\mu_t}^{\pi_t,\lambda}(\rho) \right)$$

$$- \eta_t \lambda \mathbb{E}_{s \sim d_t^\star} \left[ \mathrm{KL}(\pi_t^\star(\cdot|s) \| \pi_t(\cdot|s)) \right] - \eta_t \mathbb{E}_{s \sim d_t^\star} \left[ V_{\mu_t}^{\pi_t,\lambda}(s) \right] + \frac{\eta_t^2}{2} \|g_t\|_2^2$$

$$\leq - \eta_t \mathbb{E}_{s \sim d_t^\star, a \sim \pi_t^\star(\cdot|s)} \left[ g_t^\top \nabla_\theta \log \pi_t(a|s) - q_\mu^{\pi_t,\lambda}(s,a) \right] - \eta_t \lambda \mathbb{E}_{s \sim d_t^\star} \left[ \mathrm{KL}(\pi_t^\star(\cdot|s) \| \pi_t(\cdot|s)) \right]$$

$$- \eta_t \mathbb{E}_{s \sim d_t^\star} \left[ V_{\mu_t}^{\pi_t,\lambda}(s) \right] + \frac{\eta_t^2}{2} \|g_t\|_2^2,$$

where the last step uses the fact that $\pi_t^\star$ is the optimal (regularized) policy with respect to $\mu_t$, and thus $V_{\mu_t}^{\pi_t^\star,\lambda}(\rho) \geq V_{\mu_t}^{\pi_t,\lambda}(\rho)$. Rearranging the terms completes the proof. $\qquad\square$

## B.2 Recursive Relationship of $\mathrm{KL}(\pi_t^\star \| \pi_t)$

We define $\sigma_t^\pi \overset{\text{def}}{=} \mathbb{E}_{s \sim d_t^\star} [\mathrm{KL}(\pi_t^\star(\cdot|s) \| \pi_t(\cdot|s))] = \sum_{s \in \mathcal{S}} d_t^\star(s) \sum_{a \in \mathcal{A}} \pi_t^\star(a|s) \log \frac{\pi_t^\star(a|s)}{\pi_t(a|s)}$ as a measure of distance between $\pi_t$ and $\pi_t^\star$. Built upon Assumptions 1 and 2 and the policy improvement lemma, our next result characterizes the recursive relationship of $\sigma_t^\pi$ in terms of the approximation and statistical errors as well as the evolution of the mean-field. Such a recursion is critical to establish the convergence of the policy.

**Lemma 8.** *Under Assumptions 1 and 2, it holds that for every iteration $t \geq 0$ of Algorithm 1:*

$$\sigma_{t+1}^\pi \leq (1 - \eta_t \lambda) \sigma_t^\pi + 3 C_3 \eta_t (1 + \frac{1}{p_{\min}}) \sqrt{\varepsilon_{total}} + 2 \eta_t^2 R^2 + (1 + C_1) \cdot \kappa d_0 \|\mu_{t+1} - \mu_t\|_1,$$

*where $\kappa = \frac{2 \log |\mathcal{A}|}{1-\gamma} + \frac{1 + 2R(1-\gamma)}{\lambda(1-\gamma)}$, and $p_{\min} \geq \frac{\exp(-2R/\lambda)}{|\mathcal{A}|}$.*

*Proof.* From Lemma 1, we know that

$$\mathbb{E}_{s \sim d_t^\star} \left[ \mathrm{KL}(\pi_t^\star(\cdot|s) \| \pi_{t+1}(\cdot|s)) \right] \leq (1 - \eta_t \lambda) \sigma_t^\pi - \eta_t \mathbb{E}_{s \sim d_t^\star, a \sim \pi_t^\star(\cdot|s)} \left[ g_t^\top \nabla_\theta \log \pi_t(a|s) - q_\mu^{\pi_t,\lambda}(s,a) \right]$$

$$- \eta_t \mathbb{E}_{s \sim d_t^\star} [V_{\mu_t}^{\pi_t,\lambda}(s)] + \frac{1}{2} \eta_t^2 \|g_t\|_2^2.$$

In order to show the recursive relationship between $\sigma_t^\pi$ and $\sigma_{t+1}^\pi$, we first need to establish the relationship between $\sigma_{t+1}^\pi$ and $\mathbb{E}_{s \sim d_t^\star} [\mathrm{KL}(\pi_t^\star(\cdot|s) \| \pi_{t+1}(\cdot|s))]$, which is shown in Lemma 5 of

Appendix A. By applying the result of Lemma 5, we obtain that

$$\sigma_{t+1}^{\pi} \leq (1 - \eta_t \lambda)\sigma_t^{\pi} \underbrace{-\eta_t \mathbb{E}_{s \sim d_t^\star, a \sim \pi_t^\star(\cdot|s)} \left[ g_t^\top \nabla_\theta \log \pi_t(a|s) - q_\mu^{\pi_t, \lambda}(s, a) \right]}_{①}$$

$$\underbrace{-\eta_t \mathbb{E}_{s \sim d_t^\star}[V_{\mu_t}^{\pi_t, \lambda}(s)]}_{②} + \frac{1}{2}\eta_t^2 \|g_t\|_2^2 + (1 + C_1) \cdot \kappa d_0 \|\mu_{t+1} - \mu_t\|_1. \tag{9}$$

In the following, we upper bound each term on the RHS separately. With the compatible function approximation condition, we have that (see, e.g., [68])

$$\nabla_\theta \log \pi_\theta(a|s) = \phi_{s,a} - \sum_{a' \in \mathcal{A}} \pi_\theta(a'|s)\phi_{s,a'}.$$

We can hence rewrite ① as

$$① = -\eta_t \sum_{s,a} d_t^\star(s)\pi_t^\star(a|s) \left( \phi_{s,a}^\top g_t - \sum_{a' \in \mathcal{A}} \pi_\theta(a'|s)\phi_{s,a'}^\top g_t - q_\mu^{\pi_t, \lambda}(s, a) \right)$$

$$= -\eta_t \sum_{s,a} d_t^\star(s)\pi_t^\star(a|s) \left( \phi_{s,a}^\top g_t - \sum_{a' \in \mathcal{A}} \pi_\theta(a'|s)\phi_{s,a'}^\top g_t - Q_\mu^{\pi_t, \lambda}(s, a) + \lambda\theta_t^\top \phi_{s,a} \right)$$

$$+ \lambda\eta_t \sum_{s,a} d_t^\star(s)\pi_t^\star(a|s) \sum_{a' \in \mathcal{A}} \pi_t(a'|s)\theta_t^\top \phi_{s,a'},$$

where in the last step we used the fact that $q_\mu^{\pi_t, \lambda}(s, a) = Q_\mu^{\pi_t, \lambda}(s, a) - \lambda \log \pi_t(a|s)$ and the expression of $\log \pi_t(a|s)$. Similarly, by using the relation that

$$V_\mu^{\pi, \lambda}(s) = \mathbb{E}_{a \sim \pi(\cdot|s)}[Q_\mu^{\pi, \lambda}(s, a) - \lambda \log \pi(a|s)],$$

we can also rewrite ② as

$$② = -\eta_t \sum_{s,a} d_t^\star(s)\pi_t(a|s) \left( Q_{\mu_t}^{\pi_t, \lambda}(s, a) - \lambda \log \pi_t(a|s) \right)$$

$$= -\eta_t \sum_{s,a} d_t^\star(s)\pi_t(a|s) \left( Q_{\mu_t}^{\pi_t, \lambda}(s, a) - \lambda\theta_t^\top \phi_{s,a} \right) - \lambda\eta_t \sum_{s,a} d_t^\star(s)\pi_t(a|s) \sum_{a' \in \mathcal{A}} \pi_t(a'|s)\theta_t^\top \phi_{s,a'}.$$

Since the value of $\sum_{a' \in \mathcal{A}} \pi_t(a'|s)\theta_t^\top \phi_{s,a'}$ is independent of $a$, we have that

$$\sum_{s,a} d_t^\star(s) \left( \pi_t^\star(a|s) - \pi_t(a|s) \right) \sum_{a' \in \mathcal{A}} \pi_t(a'|s)\theta_t^\top \phi_{s,a'} = 0.$$

We can hence combine the expressions of ① and ②, and deduce that

$$①+② = -\eta_t \sum_{s,a} d_t^\star(s)\pi_t^\star(a|s) \left( \phi_{s,a}^\top g_t - \sum_{a' \in \mathcal{A}} \pi_\theta(a'|s)\phi_{s,a'}^\top g_t - Q_{\mu_t}^{\pi_t, \lambda}(s, a) + \lambda\theta_t^\top \phi_{s,a} \right)$$

$$- \eta_t \sum_{s,a} d_t^\star(s)\pi_t(a|s) \left( Q_{\mu_t}^{\pi_t, \lambda}(s, a) - \lambda\theta_t^\top \phi_{s,a} \right)$$

$$= -\eta_t \sum_{s,a} d_t^\star(s)\pi_t^\star(a|s) \left( \phi_{s,a}^\top g_t - Q_{\mu_t}^{\pi_t, \lambda}(s, a) + \lambda\theta_t^\top \phi_{s,a} \right) + \eta_t \sum_{s,a,a'} d_t^\star(s)\pi_t^\star(a|s)\pi_\theta(a'|s)\phi_{s,a'}^\top g_t$$

$$- \eta_t \sum_{s,a} d_t^\star(s)\pi_t(a|s) \left( -\phi_{s,a}^\top g_t + Q_{\mu_t}^{\pi_t, \lambda}(s, a) - \lambda\theta_t^\top \phi_{s,a} \right) - \eta_t \sum_{s,a} d_t^\star(s)\pi_t(a|s)\phi_{s,a}^\top g_t$$

$$= -\eta_t \sum_{s,a} d_t^\star(s) \left( \pi_t^\star(a|s) - \pi_t(a|s) \right) \left( \phi_{s,a}^\top g_t - Q_{\mu_t}^{\pi_t, \lambda}(s, a) + \lambda\theta_t^\top \phi_{s,a} \right),$$

where the last step holds because

$$\sum_{s,a} d_t^\star(s)\pi_t^\star(a|s) \sum_{a'} \pi_t(a'|s)\phi_{s,a'}^\top g_t - \sum_{s,a} d_t^\star(s)\pi_t(a|s)\phi_{s,a}^\top g_t$$

$$= \sum_s d_t^\star(s) \left( \sum_a \pi_t^\star(a|s) - 1 \right) \sum_{a'} \pi_t(a'|s)\phi_{s,a'}^\top g_t$$

$$= 0.$$

Using the update rule that $g_t = \hat{w}_t - \lambda\theta_t$, we further have

$$①+② = -\eta_t \sum_{s,a} d_t^\star(s) \left(\pi_t^\star(a|s) - \pi_t(a|s)\right) \left(\phi_{s,a}^\top \hat{w}_t - Q_{\mu_t}^{\pi_t,\lambda}(s,a)\right)$$

$$= -\eta_t \sum_{s,a} d_t^\star(s) \left(\pi_t^\star(a|s) - \pi_t(a|s)\right) \left(\phi_{s,a}^\top \hat{w}_t - \mathbb{E}_{a'\sim\pi_t(\cdot|s)}\left[\phi_{s,a'}^\top \hat{w}_t - Q_{\mu_t}^{\pi_t,\lambda}(s,a')\right] - Q_{\mu_t}^{\pi_t,\lambda}(s,a)\right)$$

$$= -\eta_t \sum_{s,a} d_t^\star(s) \left(\pi_t^\star(a|s) - \pi_t(a|s)\right) \left(\hat{w}_t^\top \nabla\log\pi_t(a|s) - A_{\mu_t}^{\pi_t,\lambda}(s,a)\right), \qquad (10)$$

where the second step uses the fact that

$$\sum_a \left(\pi_t^\star(a|s) - \pi_t(a|s)\right) \mathbb{E}_{a'\sim\pi_t(\cdot|s)}\left[\phi_{s,a'}^\top \hat{w}_t - Q_{\mu_t}^{\pi_t,\lambda}(s,a')\right] = 0,$$

and the third step is again due to $\nabla\log\pi_t(a|s) = \phi_{s,a} - \sum_{a'\in\mathcal{A}}\pi_t(a'|s)\phi_{s,a'}$ and the definition of $A_{\mu_t}^{\pi_t,\lambda}(s,a)$. We define

$$L_t(w) \overset{\text{def}}{=} \mathbb{E}_{s\sim d_{\mu_t}^{\pi_t}, a\sim\pi_t(\cdot|s)}\left[\left(w^\top\nabla\log\pi_t(a|s) - A_{\mu_t}^{\pi_t,\lambda}(s,a)\right)^2\right],$$

$$\text{and } \hat{L}_t(w) \overset{\text{def}}{=} \mathbb{E}_{s\sim d_{\mu_t}^{\pi_t}, a\sim\pi_t(\cdot|s)}\left[\left(w^\top\nabla\log\pi_t(a|s) - \hat{A}_t^\lambda(s,a)\right)^2\right],$$

where recall that $\hat{A}_t^\lambda(s,a) = \hat{Q}_t^\lambda(s,a) - \mathbb{E}_{a\sim\pi_t(\cdot|s)}[\hat{Q}_t^\lambda(s,a')]$ is an estimate of $A_{\mu_t}^{\pi_t,\lambda}(s,a)$ calculated using the policy evaluation oracle. From Jensen's inequality,

$$-\sum_{s,a} d_t^\star(s)\pi_t^\star(a|s)\left(\hat{w}_t^\top\nabla\log\pi_t(a|s) - A_{\mu_t}^{\pi_t,\lambda}(s,a)\right)$$

$$\leq \sum_{s,a} d_{\mu_t}^{\pi_t}(s)\pi_t(a|s)\sqrt{\left(\hat{w}_t^\top\nabla\log\pi_t(a|s) - A_{\mu_t}^{\pi_t,\lambda}(s,a)\right)^2} \cdot \frac{d_t^\star(s)}{d_{\mu_t}^{\pi_t}(s)} \cdot \frac{\pi_t^\star(a|s)}{\pi_t(a|s)}$$

$$\leq \frac{C_3}{p_{\min}}\sqrt{L_t(\hat{w}_t)}, \qquad (11)$$

where the last step uses Assumption 2 and the definition of $L_t(w)$. Recall that the policy evaluation oracle satisfies

$$\mathbb{E}\left[\left(\hat{q}_t^\lambda(s,a) - q_{\mu_t}^{\pi_t,\lambda}(s,a)\right)^2\right] \leq \varepsilon_{\text{critic}}, \forall(s,a)\in\mathcal{S}\times\mathcal{A},$$

which immediately implies that

$$\mathbb{E}\left[\left(\hat{A}_t^\lambda(s,a) - A_{\mu_t}^{\pi_t,\lambda}(s,a)\right)^2\right] \leq \varepsilon_{\text{critic}}, \forall(s,a)\in\mathcal{S}\times\mathcal{A}.$$

Let $w_t^\star = \text{argmin}_{w\in\mathbb{R}^d} L_t(w)$. From the simple fact that $(x+y)^2 \leq 2x^2 + 2y^2$, we have

$$\hat{L}_t(w_t^\star) = \mathbb{E}_{s\sim d_{\mu_t}^{\pi_t}, a\sim\pi_t(\cdot|s)}\left[\left((w_t^\star)^\top\nabla\log\pi_t(a|s) - A_{\mu_t}^{\pi_t,\lambda}(s,a) + A_{\mu_t}^{\pi_t,\lambda}(s,a) - \hat{A}_t^\lambda(s,a)\right)^2\right]$$

$$\leq 2L_t(w_t^\star) + 2\varepsilon_{\text{critic}}.$$

A similar argument shows that

$$L_t(\hat{w}_t) \leq 2\hat{L}_t(\hat{w}_t) + 2\varepsilon_{\text{critic}}.$$

Further, the gradient estimation oracle (3) guarantees that $\hat{L}_t(\hat{w}_t) - \min_w \hat{L}_t(w) \leq \varepsilon_{\text{actor}}$, and we hence obtain

$$L_t(\hat{w}_t) \leq 2\hat{L}_t(\hat{w}_t) + 2\varepsilon_{\text{critic}} \leq 2\min_w \hat{L}_t(w) + 2\varepsilon_{\text{actor}} + 2\varepsilon_{\text{critic}}$$

$$\leq 2\hat{L}_t(w_t^\star) + 2\varepsilon_{\text{actor}} + 2\varepsilon_{\text{critic}} \leq 4L_t(w_t^\star) + 2\varepsilon_{\text{actor}} + 6\varepsilon_{\text{critic}}$$

$$\leq 4\varepsilon_{\text{approx}} + 2\varepsilon_{\text{actor}} + 6\varepsilon_{\text{critic}},$$

where the last step follows from the definition of $\varepsilon_{\text{approx}}$. Plugging the above inequality back to (11), and using the fact that $\varepsilon_{\text{total}} = \varepsilon_{\text{approx}} + \varepsilon_{\text{actor}} + \varepsilon_{\text{critic}}$, we obtain that

$$-\sum_{s,a} d_t^\star(s)\pi_t^\star(a|s)\left(\hat{w}_t^\top \nabla \log \pi_t(a|s) - A_{\mu_t}^{\pi_t,\lambda}(s,a)\right) \leq \frac{3C_3}{p_{\min}}\sqrt{\varepsilon_{\text{total}}}. \tag{12}$$

Similarly, we can also get

$$\sum_{s,a} d_t^\star(s)\pi_t(a|s)\left(w_t^\top \nabla \log \pi_t(a|s) - A_{\mu_t}^{\pi_t,\lambda}(s,a)\right) \leq 3C_3\sqrt{\varepsilon_{\text{total}}}. \tag{13}$$

Substituting (10), (12), and (13) back to (9), we obtain that

$$\sigma_{t+1}^\pi \leq (1 - \eta_t\lambda)\sigma_t^\pi + 3C_3\eta_t(1 + \frac{1}{p_{\min}})\sqrt{\varepsilon_{\text{total}}} + \frac{1}{2}\eta_t^2 \|g_t\|_2^2 + (1 + C_1)\cdot\kappa d_0 \|\mu_{t+1} - \mu_t\|_1$$

$$\leq (1 - \eta_t\lambda)\sigma_t^\pi + 3C_3\eta_t(1 + \frac{1}{p_{\min}})\sqrt{\varepsilon_{\text{total}}} + 2\eta_t^2 R^2 + (1 + C_1)\cdot\kappa d_0 \|\mu_{t+1} - \mu_t\|_1,$$

where the second step holds because $\|g_t\|_2 \leq \|w_t\|_2 + \lambda \|\theta_t\|_2 \leq 2R$ due to Lemma 2.. $\qquad\square$

### B.3 Proof for Theorem 1

*Proof.* First, we know from Lemma 8 that

$$\sigma_{t+1}^\pi \leq (1 - \eta_t\lambda)\sigma_t^\pi + 3C_3\eta_t(1 + \frac{1}{p_{\min}})\sqrt{\varepsilon_{\text{total}}} + 2\eta_t^2 R^2 + (1 + C_1)\cdot\kappa d_0 \|\mu_{t+1} - \mu_t\|_1, \tag{14}$$

where $\kappa = \frac{2\log|\mathcal{A}|}{1-\gamma} + \frac{1+2R(1-\gamma)}{\lambda(1-\gamma)}$, and $p_{\min} \geq \frac{\exp(-2R/\lambda)}{|\mathcal{A}|}$. Using the mean-field state update rule that $\mu_{t+1} = (1 - \beta_t)\mu_t + \beta_t\Gamma_2(\pi_t, \mu_t)$, we have that

$$\|\mu_{t+1} - \mu_t\|_1 = \beta_t \|\mu_t - \Gamma_2(\pi_t, \mu_t)\|_1 \leq 2\beta_t.$$

Substituting the above equation back to (14) and rearranging,

$$\sigma_t \leq \frac{1}{\eta_t\lambda}(\sigma_t - \sigma_{t+1}) + \frac{3C_3}{\lambda}(1 + \frac{1}{p_{\min}})\sqrt{\varepsilon_{\text{total}}} + \frac{2\eta_t R^2}{\lambda} + \frac{2(1 + C_1)\cdot\kappa d_0\beta_t}{\eta_t\lambda}$$

Let $\eta_t = \eta = O(T^{-2/5})/\lambda, \beta_t = \beta = O(T^{-4/5})$. Summing over $t = 0, 1, \ldots, T - 1$ leads to

$$\frac{1}{T}\sum_{t=0}^{T-1}\sigma_t^\pi \leq \frac{\sigma_0}{T\eta\lambda} + \frac{3C_3}{\lambda}(1 + \frac{1}{p_{\min}})\sqrt{\varepsilon_{\text{total}}} + \frac{2\eta R^2}{\lambda} + \frac{2(1 + C_1)\cdot\kappa d_0\beta}{\eta\lambda}$$

$$\leq \widetilde{O}\left(\frac{1}{\lambda^2 T^{2/5}} + \frac{|\mathcal{A}|\exp(1/\lambda)}{\lambda}\sqrt{\varepsilon_{\text{total}}}\right). \tag{15}$$

We can then apply the Cauchy-Schwarz inequality and Pinsker's inequality to obtain that

$$\frac{1}{T}\sum_{t=0}^{T-1} D(\pi_t, \pi_t^\star) = \mathbb{E}_\tau\left[D(\pi_\tau, \pi_\tau^\star)\right]$$

$$= \mathbb{E}_\tau \mathbb{E}_{s\sim d_\tau^\star}\left[\frac{d^\star(s)}{d_\tau^\star(s)}\cdot\|\pi_\tau^\star(\cdot|s) - \pi_\tau(\cdot|s)\|_1\right]$$

$$\leq \sqrt{\mathbb{E}_\tau \mathbb{E}_{s\sim d_t^\star}\left[\left|\frac{d^\star}{d_\tau^\star}\right|^2\right]\cdot\mathbb{E}_\tau \mathbb{E}_{s\sim d_\tau^\star}\left[\|\pi_\tau^\star(\cdot|s) - \pi_\tau(\cdot|s)\|_1^2\right]}$$

$$\leq C_2\sqrt{\mathbb{E}_\tau \mathbb{E}_{s\sim d_t^\star}\left[2\text{KL}\left(\pi_\tau^\star(\cdot|s)\|\pi_\tau(\cdot|s)\right)\right]}$$

$$\leq \widetilde{O}\left(\frac{1}{\lambda T^{1/5}} + \sqrt{\frac{|\mathcal{A}|\exp(1/\lambda)\varepsilon_{\text{total}}^{1/2}}{\lambda}}\right),$$

where the last step follows from (15). This characterizes the convergence of the policy $\pi_t$. In the following, we follow a similar analysis as in [78] and analyze the convergence behavior of the mean-field state $\mu_t$. From Lemma 7, we know that

$$\|\mu_{t+1} - \mu^\star\|_1 \leq (1 - \beta_t \bar{d}) \|\mu_t - \mu^\star\|_1 + d_2 C_2 \beta_t \sqrt{\sigma_t^\pi}, \forall t \geq 0.$$

Rearranging and using the definition that $\sigma_t^\mu = \|\mu_t - \mu^\star\|_1$, we have

$$\sigma_t^\mu \leq \frac{1}{\beta_t \bar{d}}(\sigma_t^\mu - \sigma_{t+1}^\mu) + \frac{d_2 C_2}{\bar{d}} \sqrt{\sigma_t^\pi}.$$

With $\beta_t = \beta = O(T^{-4/5})$, we sum over $t = 0, 1, \ldots, T-1$ and obtain

$$\frac{1}{T} \sum_{t=0}^{T-1} \sigma_t^\mu \leq \frac{1}{T\beta\bar{d}}(\sigma_0^\mu - \sigma_T^\mu) + \frac{d_2 C_2}{T\bar{d}} \sum_{t=0}^{T-1} \sqrt{\sigma_t^\pi}$$

$$\leq \frac{\sigma_0^\mu}{T\beta\bar{d}} + \frac{d_2 C_2}{\bar{d}} \sqrt{\frac{1}{T} \sum_{t=0}^{T-1} \sigma_t^\pi}$$

$$\leq \widetilde{O}\left( \frac{1}{T^{1/5}} + \sqrt{\frac{1}{\lambda^2 T^{2/5}} + \frac{|\mathcal{A}| \exp(1/\lambda)}{\lambda} \sqrt{\varepsilon_{\text{total}}}} \right) \tag{16}$$

$$\leq \widetilde{O}\left( \frac{1}{\lambda T^{1/5}} + \sqrt{\frac{|\mathcal{A}| \exp(1/\lambda)\varepsilon_{\text{total}}^{1/2}}{\lambda}} \right), \tag{17}$$

where the second step uses the Cauchy-Schwarz inequality, and the third step follows from (15). Finally, using the triangle inequality,

$$D(\pi_t, \pi^\star) \leq D(\pi_t, \pi_t^\star) + D(\pi_t^\star, \pi^\star)$$

$$= D(\pi_t, \pi_t^\star) + D(\Gamma_1^\lambda(\mu_t), \Gamma_1^\lambda(\mu^\star))$$

$$\leq D(\pi_t, \pi_t^\star) + d_1 \|\mu_t - \mu^\star\|_1, \tag{18}$$

where the last step holds due to Assumption 3. Combining (14), (15), and (17),

$$D\left(\pi^\star, \frac{1}{T} \sum_{t=0}^{T-1} \pi_t\right) + \left\|\mu^\star - \frac{1}{T} \sum_{t=0}^{T-1} \mu_t\right\|_1 \leq \frac{1}{T} \sum_{t=0}^{T-1} D(\pi^\star, \pi_t) + \frac{1}{T} \sum_{t=0}^{T-1} \|\mu^\star - \mu_t\|_1$$

$$\leq \frac{1}{T} \sum_{t=0}^{T-1} \left(D(\pi_t, \pi_t^\star) + d_1 \|\mu_t - \mu^\star\|_1\right) + \frac{1}{T} \sum_{t=0}^{T-1} \|\mu^\star - \mu_t\|_1$$

$$\leq \widetilde{O}\left( \frac{1}{\lambda T^{1/5}} + \sqrt{\frac{|\mathcal{A}| \exp(1/\lambda)\varepsilon_{\text{total}}^{1/2}}{\lambda}} \right).$$

This completes the proof of the theorem. $\qquad\square$

## C  Instantiation of the Oracles

Section 3 assumes access to two black-box oracles that can return relatively accurate evaluations of a policy and estimations of the policy gradient. In this appendix, following the sample-based approach in [13], we discuss possible ways that the two oracles can be instantiated using standard techniques.

We start with the policy evaluation oracle, which provides an $\varepsilon_{\text{critic}}$-accurate estimate $\hat{q}$ of the shifted Q-function $q^\pi$ given a policy $\pi$. One viable approach is to instantiate such a critic oracle using temporal difference (TD) learning with linear function approximation [10, 71], a simple and widely used iterative method for policy evaluation. Specifically, we consider the case where the shifted Q-function is approximated as $q^\pi(s, a) = \psi^\top \phi_{s,a}$, where $\phi_{s,a} \in \mathbb{R}^d$ is the $d$-dimensional feature vector, and $\psi \in \mathbb{R}^d$ is the parameter vector to be optimized. The optimal $\psi$ should minimize the mean-squared projected Bellman error. A formal description of the projected TD(0) algorithm is

---

**Algorithm 2:** Projected Temporal Difference Learning with Linear Function Approximation

---

1 **Input:** Policy $\pi$ to be evaluated;
2 Initialize $\psi_0 \leftarrow \mathbf{0}$;
3 **for** *iteration $k \leftarrow 0$ to $K-1$* **do**
4     Execute policy $\pi$ to collect sample $(s_k, a_k, r_k, s_{k+1}, a_{k+1})$;
5     $\tilde{\psi}_{k+1} \leftarrow \psi_k + \alpha_k \left( r_k - \lambda \log \pi(a_k|s_k) + \gamma \psi_k^\top \phi_{s_{k+1},a_{k+1}} - \psi_k^\top \phi_{s_k,a_k} \right) \phi_{s_k,a_k}$;
6     $\psi_{k+1} \leftarrow \operatorname{argmin}_{\psi \in \mathbb{R}^d, \|\psi\|_2 \leq B} \|\psi - \tilde{\psi}_{k+1}\|_2^2$;
7 **Output:** $\hat{q}(s,a) = \frac{1}{K} \sum_{k=0}^{K-1} \phi_{s,a}^\top \psi_k$.

---

---

**Algorithm 3:** Stochastic Gradient Descent for Gradient Estimation

---

1 **Input:** Shifted Q-function estimates $\hat{q}$ from Algorithm 2;
2 Initialize $\bar{w}_0 \leftarrow \mathbf{0}$;
3 **for** *iteration $k \leftarrow 0$ to $K-1$* **do**
4     Sample $s_k \sim d^\pi$ and $a_k \sim \pi(\cdot|s_k)$ using a sampler;
5     $\bar{w}_{k+1} \leftarrow \bar{w}_k - 2\bar{\alpha}_k \left( \bar{w}_k^\top \nabla \log \pi(a_k|s_k) - \hat{A}(s_k, a_k) \right) \nabla \log \pi(a_k|s_k)$;
6     $\bar{w}_{k+1} \leftarrow \operatorname{argmin}_{w \in \mathbb{R}^d : \|w\|_2 \leq R} \|w - \bar{w}_{k+1}\|_2^2$;
7 **Output:** $\hat{w} = \frac{1}{K} \sum_{k=0}^{K-1} \bar{w}_k$.

---

presented in Algorithm 2. It starts with an initial $\psi_0$ parameter. At each iteration $k$, it executes the given policy $\pi$, and observe a sample $O_k = (s_k, a_k, r_k, s_{k+1}, a_{k+1})$ of the current state and action, the current reward, and the next state and action. The algorithm then takes a step in the direction along the negative gradient of the squared Bellman error induced by the sample $O_k$. In the entropy-regularized case, it can be shown [13] that the negative gradient is expressed as

$$g_k = \left( r_k - \lambda \log \pi(a_k|s_k) + \gamma \psi^\top \phi_{s_{k+1},a_{k+1}} - \psi^\top \phi_{s_k,a_k} \right) \phi_{s_k,a_k}.$$

The algorithm further projects the parameter $\psi$ back to a Euclidean ball of radius $B$ to ensure that the gradient norms are uniformly bounded over time. Finally, Algorithm 2 outputs an estimate of the shifted Q-function using the averaged iterate of the parameter.

Under proper assumptions (realizability and uniform mixing of the induced Markov chain, see [10] for an extensive discussion), the finite-time convergence of projected TD learning with linear function approximation is characterized in the following proposition.

**Proposition 1.** *(Theorem 3 of [10]). Under certain regularity assumptions, Algorithm 2 with a decaying step size $\alpha_k = \frac{1}{\omega(k+1)(1-\gamma)}$ ensures that*

$$\mathbb{E}\left[ \|\hat{q} - q^\pi\|_{d^\pi \times \pi}^2 \right] \leq \widetilde{O}\left( \frac{\tau^{mix}(\alpha_K)}{K(1-\gamma)^2 \omega} \right),$$

*where $\tau^{mix}(\varepsilon)$ is the $\varepsilon$-mixing time of the induced Markov chain, and $\omega$ is the smallest eigenvalue of the steady-state feature covariance matrix $\sum_{s,a} d^\pi(s)\pi(a|s)\phi_{s,a}\phi_{s,a}^\top$.*

Therefore, in order to obtain an $\varepsilon_{\text{critic}}$-accurate estimate of the shifted Q-function in expectation, it suffices to run Algorithm 2 for $\widetilde{O}(1/\varepsilon_{\text{critic}}^2)$ iterations.

Next, we instantiate the gradient estimation oracle in Algorithm 1, which provides an $\varepsilon_{\text{actor}}$-accurate estimate $\hat{w}$ of the gradient $w$, given a policy $\pi$ and an estimated value function $\hat{A}$. Since (3) solves a standard convex optimization problem, we can simply use a stochastic gradient descent (SGD) method for the actor update, which is formally described in Algorithm 3. Specifically, we first initialize the gradient estimate as $\bar{w}_0 = \mathbf{0}$. At each iteration $k$, Algorithm 3 takes a step along the opposite direction of the gradient of loss function. The gradient is given by

$$\bar{g}_k = 2 \left( \bar{w}_k^\top \nabla \log \pi(a_k|s_k) - \hat{A}(s_k, a_k) \right) \nabla \log \pi(a_k|s_k),$$

where $(s_k, a_k)$ is drawn from the distribution $d^\pi \times \pi$ (for simplicity of notations dropped the dependence on the population distribution) using a sampler (e.g., [2]), and $\hat{A}$ is calculated from $\hat{q}$

provided by the critic. The algorithm finally averages $\bar{w}_k$ over the iterations as the output. A standard result shows that Algorithm 3 with a learning step size of $\bar{\alpha}_k = \frac{R}{Q_{\max}\sqrt{K}}$ finds the optimum at a rate of $O(1/\sqrt{K})$, where recall that $Q_{\max} = \frac{1+\gamma\lambda\log|\mathcal{A}|}{1-\gamma}$:

**Proposition 2.** *(Combining Theorem 14.8 and Lemma 14.9 of [62]). Let $f : \mathbb{R}^d \to \mathbb{R}$ be a convex function, and let $x^\star = \operatorname{argmin}_{x\in\mathbb{R}^d : \|x\|_2 \leq R} f(x)$. Assume that the gradient norm at each step is bounded by $\rho > 0$ with probability 1. Suppose the (projected) stochastic gradient descent algorithm is run for $K$ iterations with the learning step size $\bar{\alpha}_k = \frac{R}{\rho\sqrt{K}}$. Then,*

$$\mathbb{E}\left[f\left(\frac{1}{K}\sum_{k=1}^{K} x_k\right)\right] - f(x^*) \leq \frac{B\rho}{\sqrt{K}}.$$

The above result immediately implies that, in order to obtain an $\varepsilon_{\text{actor}}$-accurate gradient estimation in expectation, it suffices to run Algorithm 3 for $O(1/\varepsilon_{\text{actor}}^2)$ iterations.

## D   Simulations Setup

In Subsection 5.1, we adopt two classic mean-field game tasks from the literature, including an SIS epidemics model [16, 38], and a linear-quadratic MFG [49, 38, 12, 45]. Simulations are done in an episodic setting. In our implementation, we use the collected empirical trajectory to estimate the policy gradient and the Fisher information matrix (instead of formally calculating the state visitation distribution), which turns out to serve as accurate estimates of the true values. The mean-field states are not directly observed by the learning agent, but instead only influence the environment implicitly as a parameter of the transition and reward functions.

**SIS Epidemics Model.** The SIS task describes a toy mean-field game model for epidemics. In our simulations, we consider the same setting as has been proposed in [16]. This task has two states: susceptible (S) and infected (I). At each time step, each agent may choose between two actions: social distancing (D) or going out (U). A susceptible agent will not get infected if it practices social distancing, i.e., $P(s_{t+1} = I \mid s_t = S, a_t = D, \mu_t) = 0$. When a susceptible agent chooses to go out, it has a higher probability of becoming infected if a larger proportion of the population is infected. Specifically, the state transition is given by $P(s_{t+1} = I \mid S_t = S, A_t = U, \mu_t) = 0.9^2 \cdot \mu_t(I)$, where $\mu_t(I)$ denotes the ratio of the population that is infected at time step $t$. An infected agent has a constant probability of recovery at each step, regardless of its choice of action, i.e., $P(s_{t+1} = S \mid S_t = I, A_t = U, \mu_t) = P(s_{t+1} = S \mid S_t = I, A_t = D, \mu_t) = 0.3$. For each individual agent, both practicing social distancing and being in the infected state have an associated cost, regardless of the rest of the population. Specifically, the reward function is given by $r(s, a, \mu) = -\mathbb{1}\{s = I\} - 0.5 \cdot \mathbb{1}\{a = D\}$, where $\mathbb{1}\{\cdot\}$ is the indicator function. It is worth remarking that even though this task has only two states, the transitions are also influenced by the population distribution, which is a real-valued quantity that makes this task significantly more challenging than a simple tabular MDP.

**Linear-Quadratic MFG.** The second task we consider is a 1D linear-quadratic mean-field game. We adopt the same discrete setting as has been utilized in [49, 38], which is in turn an approximation of the classic linear-quadratic MFG formulations [12, 45]. For each individual agent, the transition function of this task is given by:

$$s_{t+1} = s_t + a_t\Delta_t + \sigma\varepsilon_t\sqrt{\Delta_t},$$

where $\Delta_t$ is the time duration, and $\varepsilon_t$ is the i.i.d noise taking values from $\{-3, \ldots, 3\}$ approximately following a normal distribution $\mathcal{N}(0, 1)$. Let $\bar{\mu}_t$ denote the empirical average of the population states at time step $t$. The reward function for each agent is then specified as:

$$r(s_t, a_t, \mu_t) = \left(-\frac{1}{2}|a_t|^2 + qa_t(\bar{\mu}_t - s_t) - \frac{\kappa}{2}|\bar{\mu}_t - s_t|^2\right)\Delta_t.$$

Intuitively, this reward function incentivizes agents to track and stay close to the mean state of the population (despite the random drift $\varepsilon_t$), but discourages agents from taking large-magnitude actions. We set the parameters as $\Delta_t = 1, \sigma = 1, q = 0.01, \kappa = 0.5$, and $|\mathcal{S}| = 25$. We do not consider terminal costs in our simulations.

**Exploitability.** We utilize the standard notion of exploitability [84, 49, 16] to measure the sub-optimality of the algorithm. Specifically, the exploitability of a policy $\pi$ is defined as

$$\mathcal{E}(\pi) = \max_{\pi^\star} V_{\mu_\pi}^{\pi^\star, \lambda}(\rho) - V_{\mu_\pi}^{\pi, \lambda}(\rho),$$

where $\mu_\pi$ is the mean-field distribution generated by following the policy $\pi$, and recall that $\rho$ is the initial state distribution. Intuitively, a higher degree of exploitability suggests that an individual agent can be much better off by deviating from the given policy. On the other hand, an exploitability of $0$ suggests that the policy $\pi$ and its induced mean-field distribution constitute a Nash equilibrium of the mean-field game.

**Hyperparameter Configuration.** In the task of SIS, we set $\beta_t = 0.01$ for NAC, and $\eta_t = 0.05$ for both NAC and DL-NAC. This corroborates our theoretical findings in Algorithm 1 that the policy should evolve at a faster rate than the mean-field estimate. The learning rate of the critic is set to $0.001$ for both algorithms. In the task of LQ, we choose $\beta_t = 0.05$ for NAC, and $\eta_t = 0.1$, and a critic learning rate of $0.001$ for both NAC and DL-NAC. We use dynamic programming and the model information to calculate the exploitabilities of the policies exactly, but our algorithms do not have access to these values as they reveal information about the underlying environment. All simulation results are averaged over 10 runs.

**"Zigzags" for Fixed-Point Iterations.** The "zigzag" fluctuations of DL-NAC in Figures 1 and 2 are due to the fact that double-loop methods update the mean-field abruptly: In each "segment" of the zigzag plot, DL-NAC fixes the population distribution and learns an approximately optimal policy with respect to it. At the end of the segment, DL-NAC abruptly updates the mean-field estimate by applying one step of the mean-field dynamics operator $\Gamma_2$ under the learned policy. Such an abrupt change in the environment dynamics nullifies the policies learned from the past, and the algorithm needs to learn a new policy from scratch. This accounts for the sharp spikes in the plots of DL-NAC. Similar patterns have also been observed in the literature [16] for other fixed-point iteration methods. Our online method hence enjoys a more smooth learning behavior than standard fixed-point iteration. It is also worth remarking that the seemingly converging behavior of DL-NAC on each zigzag segment does not imply its convergence to NE, because DL-NAC fixes the mean-field estimate for each segment and does not let it get closer to the equilibrium mean-field state. In fact, even if we run each segment of DL-NAC for a sufficiently long time, there will still be a non-zero exploitability gap due to the inherent error of the mean-field estimate.

## E   Serverless FaaS Platform Setup

**OpenWhisk Cluster Setup.** A serverless Function-as-a-Service (FaaS) platform runs functions in response to invocations (i.e., function requests) from end-users or clients. Since all serverless platforms have similar master-worker architectures, we choose to use an production-grade open-source serverless platform, OpenWhisk [22], and deploy a distributed OpenWhisk cluster on IBM Cloud with 22 VMs in us-south-2. OpenWhisk manages the infrastructure, servers and scaling using Docker containers. Figure 4 shows the architecture of a distributed OpenWhisk cluster and how RL agents work with the OpenWhisk cluster to manage resources of each function. The OpenWhisk cluster that we deploy consists of one master node and 21 worker nodes. The master node runs the API gateway (labeled as ❶), FaaS controller (❷), data store (❸), and other management modules. Each of the worker nodes (labeled as ❹) runs the function containers. All nodes have 8 cores and 16-32GB RAM, running Ubuntu 20.04 LTS. There is no interference from external jobs. We run the workload generator [60] and the RL agents from two separate nodes in the same cluster and use FaaSProfiler [60] to trace requests to measure function latencies.

**Serverless Function Workflow.** The FaaS controller creates function containers, allocates CPU and RAM for each function container, and assigns the containers to worker nodes. When end-user requests arrive via the API gateway, the controller distributes the requests to worker nodes. If the function code exists on the worker node, the worker node will execute the function after it receives a request and the execution results are written to the data store; otherwise, the worker node will first pull function code from the data store before function execution. A container is evicted after an idle timeout of 10 minutes (the default value set in OpenWhisk).

**Serverless Workloads.** We select diverse function benchmarks from widely used open-source FaaS benchmark suites [15, 60, 83][4]. These benchmarks include web applications (`HTML-Gen`, `Uploader`), machine learning model serving (`Sentiment-Analysis`,`Image-Inference`), multimedia applications (`Image-Resize`, `Compression`), scientific functions (`Primes`, `PageRank`, `Graph-BFT`, `Graph-MST`), and utility functions (`Base64`, `Markdown2HTML`). The basic description of each function benchmark is listed here:

i. `Base64`: Encode and decode an input string with the Base64 algorithm.

ii. `Primes`: Find the list of prime numbers less than $10^7$.

iii. `Markdown2HTML`: Render a Base64 uploaded text string as HTML.

iv. `Sentiment-Analysis`: Generate a sentiment analysis score for the input text.

v. `Image-Resize`: Resize the input Base64-coded image with new sizes.

vi. `HTML-Gen`: Generate HTML files randomly from templates.

vii. `Uploader`: Upload a file from a given URL to Cloud storage.

viii. `Compression`: Compress given images and upload to Cloud storage.

ix. `Image-Inference`: Image recognition on a given image with a pre-trained ResNet-50 model.

x. `Page-Rank`: Calculates the Google PageRank for a specified graph.

xi. `Graph-BFT`: Traverse the given graph with breath-first search.

xii. `Graph-MST`: Generate the minimum spanning tree given a graph.

These function benchmarks have different runtime behaviors and resource demands in terms of CPU, memory, and I/O utilization. For example, `Image-Resize` and `Image-Inference` are computation-intensive functions; `Base64` and `Markdown2HTML` are memory-intensive functions; `Uploader` and `Compression` are I/O-bound functions; `Page-Rank` and `Graph-BFT/MST` are data-intensive functions (cpu- and memory-intensive). The functions are written in either Python or Java. Function-latency-based QoS objectives are defined on a per-function basis. In our experiments, we follow the common practice and use the 99th percentile latency when running in isolation on the serverless platform with 15% relaxation as the QoS latency. To drive the benchmarks, we sample and replay the function invocations from Azure function traces [61][5].

**NAC Implementation.** In the implementation of NAC-Linear and NAC-NN, due to the complexity of computing the Fisher information matrix in a large-scale environment, we use a standard gradient descent method with adaptive KL divergence penalty to approximate the policy update step, which leads to a similar procedure as the Proximal Policy Optimization algorithm [59]. We set the learning rate for both the actor and the critic networks to $3 \times 10^{-4}$. The discount factor is set to 0.99. NAC-NN has one hidden layer that consists of 64 hidden units. We set the mini-batch size and number of SGD epochs to be both 5. The reward coefficient $\alpha$ is set to 0.3, which results in the highest reward after convergence in our sensitivity study.

**Comparison Baselines.** We compare our approach with a heuristics-based approach ENSURE [67] and OpenWhisk's original resource manager. ENSURE allocates $R + c\sqrt{R}$ containers to a function with function request arrival rate $R$, scales the resources within a worker node based on a latency degradation threshold, and scales the number of worker nodes based on a memory capacity threshold. OpenWhisk sets CPU shares for each container proportional to its requested memory capacity and tries to place as many containers as possible on the same worker node to maximize the utilization. We do not include the comparison results with single-agent RL algorithms proposed for resource management (e.g., FIRM [51], MIRAS [81], and FaaSRank [82]) because these solutions are proposed under different assumptions from ours and it is hardly possible to make fair comparisons. Specially, single-agent RL solutions typically assume that the agent is in an isolated environment where there is only the application that the agent manages, but do not address the competitions for shared resources in a cluster. In fact, recent works [53, 52] have shown that applying single-agent RL algorithms to the multi-agent domain leads to severe performance degradation due to the environment's non-stationarity, which makes the single-agent solution even worse than the heuristics-based baseline that we choose.

---

[4]The benchmark FaaSProfiler uses MIT License; [15] uses BSD 3-Clause License; [83] uses Mulan Permissive Software License.

[5]The dataset [61] uses the Creative602 Commons Attribution 4.0 International Public License