# OpenReview forum: "A Mean-Field Game Approach to Cloud Resource Management with Function Approximation"
_NeurIPS.cc/2022/Conference — NeurIPS 2022 Accept_

### Official Review · Reviewer_CQL5 · 2022-07-09

**Rating:** 6
**Confidence:** 2
**Soundness:** 2 fair
**Presentation:** 2 fair
**Contribution:** 3 good

**Summary:**

This paper aims to solve the problem of improving latency and resource utilization in serverless platforms. The paper proposes a mean-field game approach where they design a natural actor-critic (NAC) learning paradigm for MFGs with function approximation.  The evaluation shows the convergence of their approach, and shows some latency and resource efficiency improvement on the OpenWhisk platform.

Note: I am not an expert on the mean-field game theory side so I am not sure about the theoretical contribution, but I am confident in my evaluation of the serverless system side. I appreciate the authors’ effort to try to formalize the resource allocation problem in serverless.


**Questions:**

1. The majority of the paper assumes homogeneous agents, and the theorems and proofs are all based on this assumption. However, the evaluation on OpenWhisk assumes heterogeneous agents. Could you provide more details on how did you adapt the algorithms for real deployment? And are those proofs still valid for heterogeneous agents? Please clarify.
2. It is unclear in the paper why this MFG approach is uniquely tailored for serverless. The theorems did not include any assumptions from the serverless system.
3. This paper only considers a very coarse-grained resource allocation type – vertically adding CPUs or memory, or horizontally adding containers. How does it scale to many different resource allocation knobs such as CPU frequency, memory bandwidth, I/O bandwidth, etc?
4. The reward function in Line 332 seems to be constructed naively by averaging CPU and memory utilizations, could you clarify that?
5. How does the resource allocation part interact with the scheduler? Do you use round-robin or least loaded worker? How do you decide which functions are co-located in one worker (function placement issue)? Those are unclear in the paper.
6. Usually resource allocation needs some time (seconds to minutes) to take effect, so the reward would only be observed after a certain delay. Therefore, resource allocation needs to be proactive, predicting what will happen next. However, the algorithm does not seem to consider such delay.
7. In prior work such as FIRM, they consider a workflow of serverless functions and incorporate dependencies between functions. How does MFG work considering dependencies?
8. Though the authors claim that scalability is the main issue of resource allocation in the cloud the paper does not provide proof or evaluate the scalability of their approach. In the evaluation, only 50 functions are tested which indicates only 50 agents are running concurrently. But the production environment usually serves thousands of different functions simultaneously.


**Limitations:**

The authors discussed some limitations and potential social network impacts in the paper. I found some limitations after reading the paper (mentioned in the above Questions section).

**Strengths And Weaknesses:**

Strengths
1. Interesting approach to map resource allocation (or autoscaling) problem in serverless to MFGs, and the authors try to solve the system challenge in a theoretica way.
2. The paper applies their approach to a real system (OpenWhisk).

Weaknesses
1. The mapping between the idealized theoretical setup and the real system is a bit vague. E.g., the major part considers homogeneous agents and only the OpenWhisk experiments consider heterogeneous agents.
2. The paper does not evaluate or prove the scalability of the proposed algorithm.

---

> ### Author Response · Authors · 2022-08-02
> **Responses to Review CQL5 (1/3)**
>
> We thank the reviewer for the valuable feedback, especially the suggestions on improving the scope of our work from multiple system design perspectives. Our detailed responses are as follows. We will also update the paper accordingly to make these points clear.
>
> 1. Since multiple existing works have performed the extension to the heterogeneous setting (see references [46, 69, 22]), we have skipped these discussions given the space limitations. Intuitively, we will need to apply our existing proofs to each type of agent, and the main results will still hold as long as there are a finite number of agent types. In the actual implementation, the only difference is that we will select a representative agent for each type of agent in the heterogeneous setting, instead of only selecting a single representative agent as in the homogenous setting. The type classification in resource management is essentially based on the function’s sensitivity to each type of resource (e.g., compute-intensive, memory-intensive, and I/O-intensive). The rest of the implementation remains unchanged.
>
> 2. In serverless computing, the cloud provider controls the resource allocation to user-registered functions. As mentioned in Section 1 paragraph 1, each function is associated with user-defined latency QoS which is the performance objective that the user wants to optimize; the provider aims to increase resource utilization by managing the resources (optimally) shared by each function without violating each QoS. As functions from self-interested users compete for shared resources in the cluster,  the problem serves as a strong and natural motivating example for our MFG formulation and solution. However, we would like to mention that even though our work is motivated by the resource management problem in cloud serverless platforms, for the sake of generality, we have aimed to make the MFG/methodology part generic enough to potentially cover a wide range of applications that can be formulated with RL, including serverless computing in particular. For example, RL has been applied in systems and networking areas such as ABRL [a], Pensieve [b], Decima [c], and many others (Park [d]). Our methodology part is intended to provide a general framework that addresses the scalability challenges for multi-agent scenarios. MFG is a suitable model to circumvent the scalability challenge by approximating the finite-agent game with an infinite-population limit. To demonstrate the effectiveness of our framework, we used serverless FaaS as one specific example, implemented the system, and evaluated the function performance managed by MFG agents. In terms of the assumption from QoS-driven serverless resource management, we have tailored the RL action, state, and reward function to better learn an optimal resource allocation policy and facilitate RL training. We would like to refer the reviewer to Appendix E for a more detailed discussion of the problem formulation under the RL pipeline, which was unfortunately not included in the main text due to space limitations.
>
>     Note that since our work is a general MFG framework independent of the underlying RL formulation so long as fitting into a sequential decision-making process, one only needs to change the RL formulation (i.e., state, action, and reward function) to apply to other RL for systems scenarios. We thank the reviewer for raising this question. We will add the explanation from this comment to the next version of our paper to establish a clearer connection between the background and our solution.
>
> 3. For vertical scaling, we only consider adjusting the number of CPU cores and the memory capacity because major commercial serverless FaaS platforms (e.g., AWS Lambda, Azure Functions, IBM Cloud Functions) only have the two as possible vertical scaling options, to the best of our knowledge. In addition, according to [55], serverless functions are not sensitive to lower-level resources such as LLC and memory bandwidth, which is also consistent with our findings during experiments. Therefore, we have chosen to include only CPU cores and memory capacity as the vertical scaling target. However, our NAC framework is applicable to more fine-granular resource allocation knobs (as mentioned in #2 by tuning the action and state space as well as the reward function).

---

> > ### Author Response · Authors · 2022-08-02
> > **Responses to Review CQL5 (2/3)**
> >
> > 4. The reward function consists of two main parts (which are also the two main objectives): one for serverless function performance (meeting QoS latency corresponds to higher rewards) and the other one for function container resource utilization (higher utilization corresponds to higher rewards). For each function, we use the average utilization across all function containers. We use the average because the metric that cloud providers are most interested in is the average resource utilization (see Google’s Datacenter as a Computer [e]). In a nutshell, the design principle of the reward function is to align with the optimization problem’s objective (which is the QoS performance and the average resource utilization).
> > 5. The control plane of a serverless platform such as OpenWhisk consists of a resource allocator (responsible for allocating resources for each container and scaling the number of containers for a function), and a request scheduler (responsible for assigning requests to any worker/container, load balancing, and admission control). In our case, we did not change the original request scheduling algorithm of OpenWhisk (which tends to send requests to the same worker until it is full because it tries to pack more load to one single server). Our approach is orthogonal/agnostic to the request scheduling algorithm: The action from the RL agent directly goes to the resource allocator to adjust the vertical and horizontal scalings; and the request scheduler picks a worker and allocated function instance to send the request based on its own static algorithm.
> > 6. Resource allocation actions from each agent do need some time to be executed/actuated by the resource allocator. In our experiments, we observed that the average latency for horizontal scaling is at O(100ms) with the p99 being around 280ms and the average latency for vertical scaling is up to 30ms (by directing writing into the cgroup resource control configuration file). Each RL step, then, spans from the time the action is generated to the time the action is executed and measurement is taken. That being said, the delay is still there because the action is not immediately applied to the environment (and this problem is a limitation for RL in general which is out of the scope of this paper). Thus, in that sense, we did not explicitly model the delay, and the experimental results illustrate how robust the learned policies are to the delay. In addition, we did not do proactive scaling (same as the single-agent RL approach FIRM) because that requires complicated time series forecasting about the client workload.
> > 7. In this work, we do not explicitly consider function dependencies. Since our approach is a reactive one that automatically scales based on the observed measurements, function dependency is instead indirectly addressed through the measured performance and utilization metrics. To explicitly model function dependencies, our framework can be potentially applied to FIRM (which does critical component localization first based on dependencies), and we consider FIRM’s critical component localization as an orthogonal and complementary work to ours. In addition, according to the paper from ATC 2020 [56], almost 60% of the workloads contain only a single function, and thus only focusing on single-function applications already accounts for a large portion of cloud workloads.

---

> > > ### Author Response · Authors · 2022-08-02
> > > **Responses to Review CQL5 (3/3)**
> > >
> > > 8. Note that given that the serverless platform architecture is a centralized model where a central manager controls all workers, it naturally cannot scale beyond a large number of servers. The central manager is usually the bottleneck for scalability. In addition, we did perform a larger-scale experiment with 120 functions and found that the percentage difference of the p99 latency for each function between the 20- and 120-agent settings is smaller than 1.9%. We have also found that as the number of agents increases, the OpenWhisk controller is not able to handle higher throughput, and thus we did not further evaluate against larger-scale settings (also due to limited time, budget, and compute resources). In reality, the large-scale clusters are managed hierarchically.  The most common way in cloud datacenters is to use a two-tier model where a large cluster is divided into a couple of sub-clusters (see [f][g][h][i][59] below). In such a model, one serverless platform can be deployed in one system pool in a sub-cluster. NAC can then be applied to each system pool/sub-cluster and interact with the central manager of the system pool for action actuation. We argue that a two-tier model could potentially solve the scalability issue (as we mentioned earlier). On the other hand, we found that a Non-MFG MARL solution such as MADDPG [j] is already not able to scale beyond 7 or 8 agents in practice (the convergence time substantially exceeds the same training budget compared to our solution). We thank the reviewer for raising this question and we will include the explanations from this comment in the discussion of scaling beyond the serverless platform’s centralized model in the Appendix.
> > >
> > >     As a final remark from a theoretical perspective, a well-known result from the MFG literature (see, e.g., [49]) says that the mean-field approximation error goes down (in the order of $1/\sqrt{N}$ where $N$ is the number of agents) as the number of agents increases. In this sense, a larger-scale system will make the MFG approach even more likely to be applicable.
> > >
> > > References:
> > >
> > > [a] Real-world Video Adaptation with Reinforcement Learning. H. Mao, S. Chen, D. Dimmery, S. Singh, D. Blaisdell, Y. Tian, M. Alizadeh, E. Bakshy. ICML 2019.
> > >
> > > [b] Neural Adaptive Video Streaming with Pensieve. H. Mao, R. Netravali, M. Alizadeh. ACM SIGCOMM 2017.
> > >
> > > [c] Learning Scheduling Algorithms for Data Processing Clusters. H. Mao, M. Schwarzkopf, S. Venkatakrishnan, Z. Meng, M. Alizadeh. SIGCOMM 2019.
> > >
> > > [d] Park: An Open Platform for Learning Augmented Computer Systems. H. Mao, P. Negi, A. Narayan, et al. NeurIPS 2019.
> > >
> > > [e] The datacenter as a computer: An introduction to the design of warehouse-scale machines. Barroso, Luiz André, Jimmy Clidaras, and Urs Hölzle. Synthesis lectures on computer architecture 8.3 (2013): 1-154.
> > >
> > > [f] Hydra: a federated resource manager for data-center scale analytics. Curino, C., Krishnan, S., Karanasos, K., Rao, S., et al. NSDI 2019.
> > >
> > > [g] Omega: Flexible, Scalable Schedulers for Large Compute Clusters. Schwarzkopf, M., Konwinski, A., Abd-El-Malek, M. and Wilkes, J. EuroSys 2013.
> > >
> > > [h] Apache Hadoop YARN: Yet Another Resource Negotiator. Vavilapalli, V.K., Murthy, A.C., Douglas, C., Agarwal, S., Konar, M., Evans, R., et al. SoCC 2013.
> > >
> > > [i] Large-scale Cluster Management at Google with Borg. Verma, A., Pedrosa, L., Korupolu, M., Oppenheimer, D., Tune, E. and Wilkes, J. EuroSys 2015.
> > >
> > > [j] Multi-Agent Actor-Critic for Mixed Cooperative-Competitive Environments. Ryan Lowe, Yi Wu, Aviv Tamar, Jean Harb, Pieter Abbeel, Igor Mordatch. NIPS 2017

---

> > > > ### Comment · Reviewer_CQL5 · 2022-08-04
> > > > **Reply to Paper3863 Authors**
> > > >
> > > > Thanks for answering my questions in such detail. I have adjusted the score accordingly.
> > > > Here are some additional comments.
> > > >
> > > > 5. I think resource allocation and scheduling are highly related. Because the function placement ultimately determines the resource sharing quality. For example, if you pack many containers on one worker, you won't be able to allocate more resources to satisfy SLO because the worker's total amount of resources is limited. In other words, a system needs both a good resource allocator and a good scheduler. It would be interesting to see some discussions in the paper.
> > > >
> > > > 8. Just a minor comment regarding "the serverless platform architecture is a centralized model where a central manager controls all workers". It is probably true just for OpenWhisk, which does not have a scalable architecture. By contrast, commercial platforms like AWS Lambda or Azure Functions can scale to thousands of concurrent functions.

---

> > > > > ### Author Response · Authors · 2022-08-05
> > > > > **Response to Reviewer CQL5**
> > > > >
> > > > > Thanks for the additional comments and we appreciate your time for discussing in more detail!
> > > > >
> > > > > 5. We do agree that resource allocation and scheduling are highly related. When we fix the function placement and scheduling algorithm, the resource allocation policy learned by the RL agent will be the optimal policy given the function placement/scheduling “state”. In another word, the objective of the resource allocator is mainly for guaranteeing QoS under the function placement/scheduling framework, which is the scope of the RL formulation.
> > > > > Note that we are not claiming that it is the optimal resource allocation policy at **any** time because as you also mentioned, the policy should naturally be a joint effort from both the function placement/scheduling control plane and the resource allocation control plane. In addition, how quickly resource requests issued from the framework can be guaranteed is determined by the underlying cluster autoscalers/schedulers on a container orchestration platform (like Kubernetes). And indeed, the coordination between serverless resource allocation and the underlying system scheduling would be an interesting question to study.
> > > > >
> > > > > 6. Thanks for the comment! We do want to clarify two concepts: (1) serverless computing platform; and (2) cluster management or container orchestration platform (like Borg / Kubernetes / Twine / YAWN). To the best of our knowledge, the serverless computing platform is still using a master-worker architecture in any open-sourced or commercial serverless platform. However, the reason why “commercial platforms like AWS Lambda or Azure Functions can scale to thousands of concurrent functions” is because their underlying cluster management platform is hierarchical as we mentioned earlier. Their way of scaling is through a hierarchical design (but for cluster management like Kubernetes not for the serverless platform). In a hierarchical cluster management platform, to scale to thousands of concurrent functions, each small/medium-sized sub-cluster/system pool can be used to deploy one serverless platform (different sets of functions run on different clusters), on which our framework can be applied.
> > > > >
> > > > > We also want to mention that the experiments we did (including both simulation-based experiments and the real experiments on a serverless computing system) serve as a validation of the theory. With these validation results, our MFG formulation and NAC framework (with algorithm proved theoretically) have the potential to be applied to a wide range of problems (e.g., RL for congestion control, video streaming, power management, etc.) and open the door to welcome more systems research based on the theory, namely the convergence results and the approximation errors would decrease as the number of agents increases.

---

### Official Review · Reviewer_6cfc · 2022-07-10

**Rating:** 6
**Confidence:** 3
**Soundness:** 3 good
**Presentation:** 2 fair
**Contribution:** 2 fair

**Summary:**

The paper presents a mean field game approach to the many selfish agent problem. The paper contextualises this around the resource management problem for FaaS before moving on to a theoretical evaluation of the problem and finishes with experimental results.

**Questions:**

The paper is in general well written. No small questions.

**Limitations:**

This was not discussed.

**Strengths And Weaknesses:**

The paper presents a clear introduction to the FaaS problem along with a detailed mathematical analysis of the MFG solution.

However, the paper is very dense and feels like two papers rammed into one. The introduction is all about the Cloud and the FaaS problem. The next section of the paper is a dense mathematical work on the MFG problem - with no mention of the FaaS problem. The FaaS only returns as one of the examples in the results section. It would have probably been better to have couched this paper as a fully theoretical paper and just keep the FaaS as one of the results examples. This would have given more space for covering the 'middle paper'. This could have helped in making it clear what the 'NeurIPS' element of this work was.

---

> ### Author Response · Authors · 2022-08-02
> **Responses to Review 6cfc**
>
> We thank the reviewer for the insightful feedback. Regarding the (potential) disjointness between the FaaS problem and the MFG formulation, we would like to mention that serverless computing is a natural application scenario of the MFG formulation, and serves as a strong motivation to study the convergence properties of learning algorithms in MFGs. For the sake of generality, we aimed to make the MFG framework generic enough to potentially cover a wide range of applications that can be formulated with RL (e.g., congestion control, video streaming bitrate adjusting, and power management), including serverless computing in particular. To demonstrate the effectiveness of our framework, we used serverless FaaS as one specific example, implemented the system, and evaluated the function performance managed by MFG agents. To see a clearer connection between the specific serverless problem and the generic MFG formulation, we would like to refer the reviewer to Appendix E for a more detailed discussion of the problem formulation under the RL pipeline, which was unfortunately not included in the main text due to space limitations. Specifically, we have discussed the formulation of resource management in a serverless platform as a sequential decision-making problem that can be solved by the MFG/RL framework (illustrated in Fig. 6). By approximating the finite-agent game with an infinite-population limit, our solution largely resolves the scalability challenge. Our theoretical proof then establishes the finite-time convergence formally and provides support for our proposed solution. We thank the reviewer for pointing out such disconnection. We will make these points clear in the main text of the paper to ensure a smooth transition. We would also be happy to discuss further if the reviewer has other questions or comments about the paper.

---

### Official Review · Reviewer_fRko · 2022-07-14

**Rating:** 6
**Confidence:** 3
**Soundness:** 3 good
**Presentation:** 2 fair
**Contribution:** 3 good

**Summary:**

The competition and scheduling of cloud service resources is a significant research problem with economic value. Due to the emergence of new cloud service paradigms such as serverless computing, we consider the aid of multi-agent reinforcement learning (MARL) to address new management challenges. To this end, the authors design a mean-field game (MFG) approach that can efficiently manage large-scale cloud users and applications. They propose an online natural actor-critic algorithm for approximating functions in mean-field games and demonstrate the finite-time convergence of the algorithm theoretically. The study evaluates the solution's effectiveness on OpenWhisk using workloads captured from real production systems, showing advantages in scalability, latency, and resource utilization.

**Questions:**

Overall, this paper constructs a comprehensive multi-agent reinforcement learning system and demonstrates the properties of online natural actor-critic algorithms in detail. Still, the most prominent weakness is the lack of necessary problem formulation (i.e., cloud resource management). Therefore, at the very beginning of the paper, it is suggested to supplement the problem case study and formalization to formalize the resource scheduling problem into a multi-agent reinforcement learning process in the context of a serverless computing scenario. This issue involves a lot of real details and needs to be explained. For example, cloud users do not need to manage the computing resources in a serverless computing scenario, so how can self-interested users compete for resources.

**Limitations:**

Not applicable.

**Strengths And Weaknesses:**

Strengths:
- The cloud resource competition and scheduling problem has real research significance and economic value
- Comprehensive theoretical basis and proof of properties
- The efficiency advantage of the proposed solution makes it adaptable to real system scenarios

Weaknesses：
- The background and problem formalization are insufficient, resulting in the disjointed problem background and methodology
- Validation in large-scale system scenarios/cases

---

> ### Author Response · Authors · 2022-08-02
> **Responses to Review fRko (1/2)**
>
> We thank the reviewer for the insightful feedback. Our detailed responses are as follows.
> 1. Regarding the potential disjointness between background and methodology, we would like to mention that the serverless resource management problem is a strong motivating application for our MFG framework and NAC solution but our framework could potentially be applied to other applications that can be formulated with RL. In serverless computing, the cloud provider controls the resource allocation for user-registered functions. As mentioned in Section 1 paragraph 1, each function is associated with user-defined latency QoS which is the performance objective that the user wants to optimize; the provider aims to increase resource utilization by managing the resources (optimally) shared by each function without violating each QoS. That’s why “self-interested users compete for shared resources”, to answer your specific question. To see a clearer connection between the specific serverless problem and the generic MFG formulation, we would like to refer the reviewer to Appendix E for a more detailed discussion of the problem formulation under the RL pipeline, which was unfortunately not included in the main text due to space limitations. We model the resource management in a serverless platform as a sequential decision-making problem that can be solved by the RL framework (illustrated in Fig. 6). At each step in the sequence, the RL agent (labeled as <5>) monitors the system and application conditions from both the OpenWhisk data store (labeled as <3>) and the Linux cgroups. Measurements include function-level performance statistics (i.e., tail latencies on execution time, waiting time, and cold-start time for serving function requests) and system-level resource utilization statistics (e.g., CPU utilization of function containers). These measured telemetry data are pre-processed and used to define a state, which is then mapped to a resource management decision by the RL agent. In this model, we consider both vertical and horizontal resource scaling actions (described in Sec. 5.2). The decision made by the RL agent is then passed by the horizontal and vertical scaler to the FaaS controller (<2>), and finally changes the system state and function performance, which finishes an RL state transition.
>
>
>     On the other hand, even though our work is motivated by the resource management problem in cloud serverless platforms, we believe that it is also applicable to other RL frameworks as well. For instance, RL has been applied in the systems and networking area (such as video streaming, congestion control, power management, and so on). For the sake of generality, we aimed to make the MFG framework generic enough to potentially cover a wide range of applications that can be formulated with RL. Our aim was to provide a general framework that addresses the scalability challenges for multi-agent scenarios. MFG is a suitable model to circumvent the scalability challenge by approximating the finite-agent game with an infinite-population limit. To demonstrate the effectiveness of our framework, we used serverless FaaS as one specific example, implemented the system, and evaluated the function performance managed by MFG agents. In terms of the assumption from QoS-driven serverless resource management, we have tailored the RL action, state, and reward function to better learn an optimal resource allocation policy and facilitate RL training. Note that since our work is a general MFG framework independent of the underlying RL formulation so long as fitting into a sequential decision-making process, one only needs to change the RL formulation (i.e., state, action, and reward function) to apply to other RL for systems scenarios. We thank the reviewer for raising this question and we will add the explanation from this comment to the next version of our paper to build a clearer connection from the background to our solution.

---

> > ### Author Response · Authors · 2022-08-02
> > **Responses to Review fRko (2/2)**
> >
> > 2. Regarding the validation in large-scale system cases, note that given the serverless platform architecture is a centralized model where a central manager controls all workers, it naturally cannot scale beyond a large number of servers. The central manager is usually the bottleneck for scalability. In addition, we did perform a larger-scale experiment with 120 functions and we found that the percentage difference of the p99 latency for each function between the 20- and 120-agent settings is smaller than 1.9%. We also found that as the number of agents increases, the OpenWhisk controller is not able to handle higher throughput and thus we did not further evaluate against larger-scale settings (also due to limited time, budget, and compute resources). The most common way to deal with scalability in cloud datacenters is to use a two-tier model where a large cluster is divided into a couple of sub-clusters (see [a][b][c][d][59]). In such a model, one serverless platform can be deployed in one system pool in a sub-cluster. NAC can then be applied to each system pool and interact with the central manager of the system pool for action actuation. We argue that a two-tier model could potentially solve the scalability issue (as we mentioned earlier). We thank the reviewer for raising this question, and we will include the explanations from this comment in the discussion of scaling beyond the serverless platform’s centralized model in the Appendix.
> >
> >
> > References:
> >
> > [a] Hydra: a federated resource manager for data-center scale analytics. Curino, C., Krishnan, S., Karanasos, K., Rao, S., et al. NSDI 2019.
> >
> > [b] Omega: Flexible, Scalable Schedulers for Large Compute Clusters. Schwarzkopf, M., Konwinski, A., Abd-El-Malek, M. and Wilkes, J. EuroSys 2013.
> >
> > [c] Apache Hadoop YARN: Yet Another Resource Negotiator. Vavilapalli, V.K., Murthy, A.C., Douglas, C., Agarwal, S., Konar, M., Evans, R., et al. SoCC 2013.
> >
> > [d] Large-scale Cluster Management at Google with Borg. Verma, A., Pedrosa, L., Korupolu, M., Oppenheimer, D., Tune, E. and Wilkes, J. EuroSys 2015.

---

> > > ### Comment · Reviewer_fRko · 2022-08-08
> > > **Reply to Paper3863 Authors**
> > >
> > > Thanks to the authors for their detailed reply. I expected the paper to discuss cloud resource management in more detail since the title is "A Mean-Field Game Approach to Cloud Resource Management with ***". However, putting a lot of scenario descriptions in the appendix does not help the reader to understand how to formulate the problem with RL and why the MFG framework and NAC can solve it well, especially for readers from the industry.
> > >
> > > Alternatively, the paper could claim to build MFG frameworks and NAC solutions for ALL RL-compatible application problems from a more general perspective. In this way, the contribution of this paper will be more valuable.
> > >
> > > Anyway, this is just a writing issue. I believe the proposed solution has good potential to be applied to other scenarios. Therefore, the current score is appropriate.

---

### Official Review · Reviewer_BAei · 2022-07-14

**Rating:** 6
**Confidence:** 3
**Soundness:** 3 good
**Presentation:** 3 good
**Contribution:** 3 good

**Summary:**

The paper proposes a scalable mean-field game (MFG) approach to the problem of cloud resource management. It introduces an online natural actor-critic (NAC) algorithm for MFGs, which uses function approximation and lets the mean-field state naturally evolve as the agents learn. The authors establish finite-time convergence of NAC with linear function approximation and softmax parameterization, but also implement neural-net based function approximation. Experimental results on open-source serverless platform OpenWhisk with real-world workloads from production traces demonstrate that NAC is scalable to a large number of agents and significantly outperforms other common baselines.

**Questions:**

1. In terms of theory, what are the main differences of this work with the NAC setting with linear function approximation by (Cayci, He, and Srikant, 2021)? I understand that the setting is a bit different, because this work focuses on mean-field approximation for a multi-agent setting, whereas the cited work is purely single-agent. On the other hand, the mean-field approach effectively transforms the problem into a single-agent one. What are the additional innovations/challenges (particularly in terms of theory and proofs) of this work compared to the work by (Cayci, He, and Srikant, 2021)?
2. Why have the authors not included any RL-based baseline (except for the double-loop version of NAC)? Multi-agent RL-based approaches without the mean-field assumption may suffer from limited scalability, so the authors could show the benefit of MFG NAC in a rather basic setting. Is it also possible to compare single-agent RL-based approaches (e.g., FIRM) to the proposed framework, e.g., by looking at the problem from the perspective of the single agent?
3. In Equation (1), right in the middle do the authors perhaps mean V^{\pi,\lambda}_{\mu^*}(\rho) instead of V^{\pi}_{\mu^*}(\rho)? Shouldn't the \lambda appear somewhere?

---AFTER REBUTTAL---
The authors addressed my concerns; I am therefore willing to raise my recommendation to Weak Accept.

**Limitations:**

The authors have addressed the limitations of their work.

**Strengths And Weaknesses:**

Strengths
1. The paper is well written. Even the theory parts that are more involved are not hard to follow and the authors provide adequate refences and clarifications to explain the different concepts. I found this very helpful, because the NAC algorithm contains a number of steps such as policy evaluation, gradient estimation, mean-field update and policy update.
2. The problem that motivates this work is an important real-world problem. Furthermore, a MFG approach is a quite reasonable design choice for this problem, in particular for large numbers of users where collective user behavior can be summarized by a population distribution, and a common policy can be applied to all agents.
3. I was unable to check the proofs in detail, but the proposed algorithm and the ingredients therein make sense, as they follow faithfully various prior works.
4. The related work is concise but seems to cover a lot of important and relevant prior works.
5. The experiments show that the proposed NAC does not suffer from the zigzag fluctuations of the double-loop version of NAC that uses a fixed point iteration. Furthermore, experiments with real-world workloads from production traces show that NAC significantly outperforms other common baselines such as ENSURE and OpenWhisk's original resource manager.

Weaknesses
1. The novelty is not strong. Convergence of entropy-regularized natural policy gradient with linear function approximation has been already studied by (Cayci, He, and Srikant, 2021). And indeed this work builds very heavily upon this specific work (which, incidentally, is a pre-print and has not been accepted to peer-reviewed venues yet). The main difference is that NAC is now applied to a MFG setting, where the main solution concept is that of a Nash Equilibrium. Still, due to the mean-field approximation, the agent essentially faces a single-agent policy optimization problem, and the game-theoretic setting roughly becomes equivalent to a single-agent Markov decision process. That said, I do not claim that the two works are identical, since the deal with a different setting; however, they do overlap significantly.
2. In the experimental evaluation, NAC is only compared to two other heuristic-based baselines. It would have made much more sense to also compare against other RL-based schemes. As the authors point out in the related work, there are various RL-based frameworks for scheduling or resource management (e.g., FIRM, FaaSRank, etc.). I feel that the authors would have made a much stronger point is they were able to show that the NAC algorithm can outperform other RL-based competitors (even by focusing on a simpler objective such as minimizing latency only). Even if some approaches are single-agent, the authors could also experiment multi-agent RL-based baselines to showcase the benefits of their framework (even in a basic setup to demonstrate the scalability challenges with traditional multi-agent approaches).

---

> ### Author Response · Authors · 2022-08-02
> **Responses to Review BAei (1/2)**
>
> We thank the reviewer for the valuable comments. Our detailed responses are as follows.
> 1. We appreciate the reviewer’s concern regarding the comparison with (Cayci et al., 2021) in terms of theory, but we would like to emphasize that the online (“single-loop”) nature of our MFG approach introduces an additional non-stationarity challenge (which is inescapable in multi-agent scenarios) and makes our analysis very different from a single-agent one (Cayci et al., 2021). Specifically, the reviewer is correct in that if one uses a standard “double-loop” mean-field approach, the setting will be very similar to that of (Cayci et al., 2021) because by fixing the mean-field state (population distribution), the learning agent effectively faces a single-agent problem. However, such double-loop solutions are hardly practical, since it is almost impossible to freeze the population distribution in a cloud computing system to let a representative agent learn an optimal policy. Such methods also suffer undesirable “zigzag” fluctuations as we demonstrated in our simulations. What differentiates our approach from single-agent policy optimization (Cayci et al., 2021) is the fact that we are considering a more practical online (“single-loop”) mean-field learning paradigm. In our setting, the environment (transition and reward functions) simultaneously evolves as the agents update their policies. The environmental non-stationarity hence adds another layer of complexity, because the agent needs to be aware of the impact it has on the environment when it updates its policy. To deal with this challenge (which does not exist in Cayci et al., 2021), our solution carefully balances the time scales of policy updates and mean-field estimates (in a non-asymptotic manner) to ensure that the policies are still updated in a consistent way even under environmental non-stationarities (primarily due to actions of other agents). This also explains why our convergence rate is very different from that of single-agent optimization (Cayci et al., 2021). We thank the reviewer for raising this point, and we will make this point clearer to the readers in the next version of the paper.
>
> 2. We did not present in this paper the comparison results with single-agent RL algorithms proposed for resource management (e.g., FIRM [48], MIRAS [ICDCS 2019], and FaaSRank [74]) for two main reasons. (1) Firstly, those works that proposed single-agent RL solutions have different assumptions from ours and it would not an apple-to-apple comparison. In their single-tenant isolated training environment, the agent could train to convergence and achieve optimal performance. Single-agent RL solutions like FIRM assume that the agent is in an isolated environment where there is only the application that the agent manages. However, in our case, functions from all customers compete for shared resources in a cluster and all agents are concurrently interacting with the environment. From the point of view of each agent, the environment is no longer stationary in the multi-agent domain. The single-agent RL solution is not aware of the other agents in the same environment. At the training stage, since the state transitions and rewards depend on the joint actions of all agents whose policies keep changing during the learning process, each agent enters an endless cycle of adapting to other agents. We observed that the policy learned by an agent could not converge during training. (2) Secondly, with the same training budget, we picked the last checkpoint from RL training (though not converged) and evaluated the online policy-serving performance (during the inference stage). We found that due to performance degradation when applying single-agent RL algorithms to the multi-agent domain, the single-agent RL solution is even worse than the heuristics-based approach (our baseline). This is because, at the policy-serving stage, the underlying environment could be updated (by other agents) during the time between an action is generated (by an agent) and executed by the FaaS controller. We performed a more comprehensive measurement study of applying single-agent RL solutions in multi-tenant serverless platforms in another paper but we did not include that reference to provide anonymity.

---

> > ### Author Response · Authors · 2022-08-02
> > **Responses to Review BAei (2/2)**
> >
> > 2. (Continued) Let’s also consider a concrete example where there are two agents in the environment (which used a single-agent RL algorithm to independently train to convergence), each controlling one function. Suppose that given the current state, agent-1 makes the (optimal) decision to scale up CPU shares by 256 to achieve optimality. When both agent-1 and agent-2 are present (at the same current state), agent-2 also wants to scale up CPU shares by 512 (simultaneously), which affects the final CPU share ratio for both agents. Then, the previously generated decision by agent-1 (for the single-agent case) is no longer optimal in the two-agent case. In our preliminary experiments, we have observed that the suboptimal policy (by single-agent RL algorithm) resulted in up to 14x performance degradation compared to our MARL solution in terms of the end-to-end p99 latency. We thank the reviewer for raising this question. In the next version of our paper, we will make the point clearer to the reader that the single-agent RL algorithm suffers from environment non-stationarity issues and performance degradation in multi-agent settings.
> >
> > 3. We thank the reviewer for asking this clarifying question. We would like to clarify that \lambda will not appear in Equation (1). Equation (1) is used to show that the regularized NE is a good approximation of the unregularized one. Specifically, it shows that when applying the regularized NE (\pi^\star, \mu^\star) back to the unregularized MFG, its corresponding value should be close to that of the unregularized NE in the same unregularized MFG. Since the values we compare are both defined based on the unregularized MFG, there is no superscript of \lambda in the notation of the value function V. Instead, \lambda (implicitly) appears in \pi^\star and \mu^\star, because the regularized NE (\pi^\star, \mu^\star) will be different for different levels of \lambda of regularization.

---

### Meta-Review · Area_Chair_TAez · 2022-08-24

**Recommendation:** Accept
**Confidence:** Certain

**Metareview:**

I agree with the reviewers that this is a well-written paper on an interesting application of mean-field games. The paper is a nice blend of theoretical developments and experimental evaluations. I believe that it will be well-received by the NeurIPS community and recommend acceptance.

**Award:**

No

---

### Decision · Program_Chairs · 2022-09-14

Accept